# Evidence for DNA-mediated nuclear compartmentalization distinct from phase separation

David Trombley McSwiggen[1,2], Anders S Hansen[1,2], Sheila S Teves[1,3], Hervé Marie-Nelly[1,2], Yvonne Hao[1], Alec Basil Heckert[1,2], Kayla K Umemoto[1], Claire Dugast-Darzacq[1,2], Robert Tjian[1,4]*, Xavier Darzacq[1]*

[1]Department of Molecular and Cell Biology, University of California, Berkeley, Berkeley, United States; [2]California Institute of Regenerative Medicine Center of Excellence, University of California, Berkeley, Berkeley, United States; [3]Department of Biochemistry and Molecular Biology, University of British Columbia, Vancouver, Canada; [4]Howard Hughes Medical Institute, University of California, Berkeley, Berkeley, United States

**Abstract** RNA Polymerase II (Pol II) and transcription factors form concentrated hubs in cells via multivalent protein-protein interactions, often mediated by proteins with intrinsically disordered regions. During Herpes Simplex Virus infection, viral replication compartments (RCs) efficiently enrich host Pol II into membraneless domains, reminiscent of liquid-liquid phase separation. Despite sharing several properties with phase-separated condensates, we show that RCs operate via a distinct mechanism wherein unrestricted nonspecific protein-DNA interactions efficiently outcompete host chromatin, profoundly influencing the way DNA-binding proteins explore RCs. We find that the viral genome remains largely nucleosome-free, and this increase in accessibility allows Pol II and other DNA-binding proteins to repeatedly visit nearby DNA binding sites. This anisotropic behavior creates local accumulations of protein factors despite their unrestricted diffusion across RC boundaries. Our results reveal underappreciated consequences of nonspecific DNA binding in shaping gene activity, and suggest additional roles for chromatin in modulating nuclear function and organization.
DOI: https://doi.org/10.7554/eLife.47098.001

*For correspondence:
jmlim@berkeley.edu (RT);
darzacq@berkeley.edu (XD)

## Introduction

Controlling the local concentration of molecules within cells is fundamental to living organisms, with membrane-bound organelles serving as the prototypic mechanism. In recent years, our understanding of the forces driving the formation of sub-nuclear compartments has undergone a paradigm shift. A number of studies suggest that many proteins have the ability to spontaneously form separated liquid phases in vitro (*Banani et al., 2017*), and recent work highlights the possibility that similar liquid compartments may occur in vivo (*Courchaine et al., 2016*; *Bracha et al., 2018*). Such liquid-liquid demixing (liquid-liquid phase separation, LLPS) has been proposed to be a common mechanism in sequestering specific macromolecules within a compartment, or in increasing their local concentration and thereby facilitating molecular interactions. Formation of these structures is thought to be predominantly driven by multivalent interactions mediated through intrinsically disordered regions (IDRs), or via modular binding motifs, RNA, or DNA (*Banani et al., 2017*).

These observations have generated a deeper appreciation for the diversity of mechanisms that a cell may deploy so as to locally concentrate select molecular constituents. The list of proteins—particularly nuclear proteins—that can undergo phase separation in vitro continues to grow

(*Courchaine et al., 2016*). For example, recent studies of RNA Polymerase II (Pol II) and its regulators have shown that Pol II forms dynamic hubs whose sizes depend on the number of intrinsically disordered heptad peptide repeats contained within the C-terminal domain (CTD) (*Boehning et al., 2018*), and that various CTD interacting factors may form phase-separated droplets in vitro (*Lu et al., 2018*) as well as local concentration hubs in vivo (*Chong et al., 2018*). We do not, however, fully understand the nature of the molecular forces that drive compartmentalization, and we lack compelling evidence of the functional consequences of these compartments.

Herpes Simplex Virus type 1 (HSV1) lytic infection provides an attractive model system because of its ability to form nuclear compartments de novo. HSV1 hijacks its host's transcription machinery during lytic infection (*Rice et al., 1994*), transcribing its genome in three waves: immediate early, early, and late, with the latter strictly occurring only after the onset of viral DNA replication (*Knipe and Cliffe, 2008*). Viral replication generates subcellular structures called replication compartments (RCs) where both viral and host factors congregate to direct replication of the viral genome, continue viral transcription, and assemble new virions (*Knipe and Cliffe, 2008*). Recent reports highlight the ability of HSV1 to hijack host Pol II such that, once late gene transcription commences, the host chromatin is largely devoid of productively transcribing Pol II, and the majority of newly synthesized mRNAs are viral in origin (*Abrisch et al., 2015*; *Rutkowski et al., 2015*). Concomitantly, RCs show a dramatic enrichment of Pol II and other nuclear factors (*Rice et al., 1994*).

Given this shift in both the sub-nuclear localization of Pol II upon infection, and its effect on the transcriptional output of an infected cell, we chose to examine the mechanism of Pol II recruitment to HSV1 RCs as a model case for the generation of new subcellular compartments. We employed a combination of imaging approaches, and complemented these with genetic, genomic, and chemical perturbation experiments while measuring Pol II behavior in infected and uninfected cells. Despite initial indications that RCs exhibit many of the macroscopic hallmarks of LLPS, we find that recruitment of Pol II and other DNA-binding proteins to RCs is achieved through a distinct compartmentalization mechanism. Pol II recruitment occurs predominantly through transient, nonspecific binding of Pol II to viral DNA. These interactions are independent of transcription initiation, relying instead on the unusual feature that the HSV1 genome is largely free of nucleosomes, and therefore hyperaccessible to DNA-binding proteins relative to host chromatin. Our findings reveal that nonspecific binding can play a key role in the recruitment and retention of Pol II during infection, and more generally in the repertoire of distinct mechanisms a cell might employ to generate membraneless compartments.

## Results

### Pol II recruitment to RCs exhibit hallmarks of liquid-liquid demixing

HSV1 replication compartments form de novo following lytic infection, making them an attractive system to dissect compartment formation at the molecular level. To determine the mechanisms leading to the hijacking of Pol II, we used a U2OS cell line in which the catalytic subunit of Pol II has been fused to HaloTag (*Boehning et al., 2018*). HSV1 infection occurs rapidly, with large RCs forming within a few hours (*Figure 1A*). Because we were most interested in the early stages of lytic infection when Pol II is actively recruited to the RC, we focused our experiments on the period between 3 hours post infection (hpi), when RCs begin to emerge, and 6 hpi when infected cells begin to display significant cytopathic effects (*Figure 1—video 1* and *2*).

In addition to Pol II, many other viral and nuclear factors re-localize to RCs (*Dembowski and DeLuca, 2015*). This redistribution of proteins is so dramatic that it can be seen as a change in the refractive index of RCs (*Figure 1A*). RCs grow and move over the course of infection (*Figure 1B*), and RCs exhibit other behaviors characteristic of liquid droplets, such as fusion (*Figure 1B*; *Figure 1—video 1* and *2*) and a spherical shape with an aspect ratio close to one (*Figure 1C*), reminiscent of interfaces subject to surface tension (*Brangwynne et al., 2011*).

Another hallmark of LLPS compartments is that they are commonly associated with enrichment in proteins with IDRs. Across all HSV1 proteins, we identified predicted IDRs based on the protein sequence (*Figure 1D*). When categorized by temporal class, the immediate early (IE) and viral tegument proteins—the two groups that are first available to the cell upon infection—had the highest fraction of predicted intrinsic disorder. Compared to a list of proteins known to undergo LLPS in

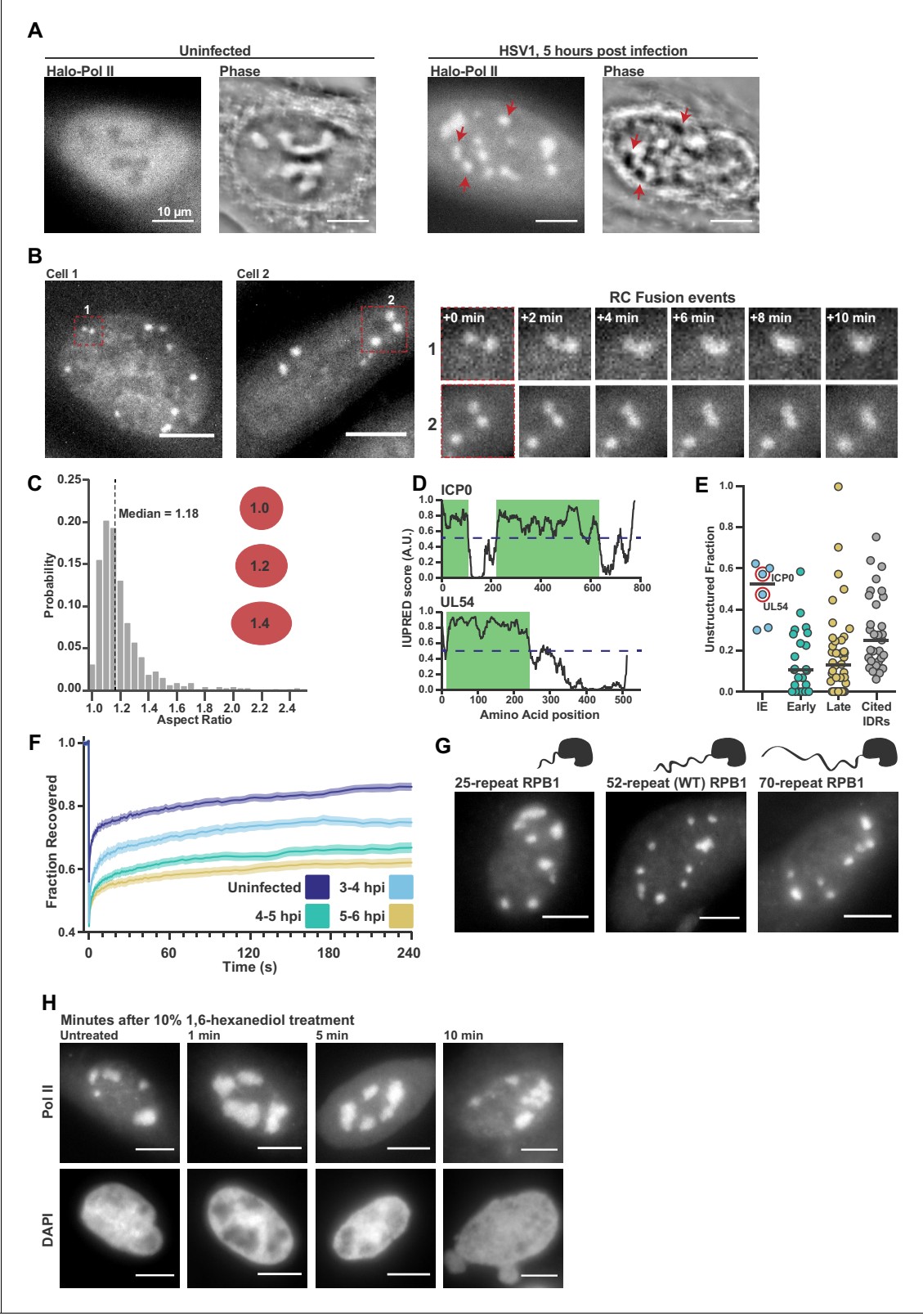

**Figure 1.** Pol II recruitment to Replication Compartments exhibits hallmarks of liquid-liquid demixing. (**A**) Representative fluorescence and phase images in uninfected and infected cells. RCs shows a different phase value compared with the surrounding nucleoplasm. Red arrows show matched examples of RCs in the two channels. (**B**) Time-lapse images of Pol II recruitment to RCs. Zoom in shows RC fusion events. See also *Figure 1—video 1* and *2*. (**C**) Aspect ratios (max diameter/min diameter) of RCs from 817 RCs in 134 cells, 3 to 6 hpi. Red ellipses provided a guide to the eye of different

*Figure 1 continued on next page*

*Figure 1 continued*

aspect ratios. (D) IUPred scores for two Immediate Early viral proteins, ICP0 and UL54, as a function of residue position. Green boxes are predicted IDRs. (E) The fraction of each protein in the viral proteome that is unstructured, separated by kinetic class. HSV1 proteins are compared to a curated list of proteins containing IDRs known to drive phase separation (Cited IDRs). (F) FRAP curves of Pol II in RCs from 3 to 4 hpi, 4–5 hpi, and 5–6 hpi (n = 24, 33, and 33), compared with uninfected cells (n = 31). Shown is the mean flanked by SEM. (G) Infected HaloTag-RPB1 cell lines with a C-terminal domain containing different numbers of heptad repeats. (H) Pol II localization 1, 5 and 10 min after 10% 1,6-hexanediol treatment. All scale bars are 10 μm. Source data for of the list of IDRs in the HSV genome as well as previously cited IDRs can be found in *Figure 1—source datas 1* and *2*, respectively.
DOI: https://doi.org/10.7554/eLife.47098.002

The following video, source data, and figure supplement are available for figure 1:

**Source data 1.** List of putative IDRs in the HSV1 genome identified by IUPred.
DOI: https://doi.org/10.7554/eLife.47098.004

**Source data 2.** List of proteins reported to undergo phase separation.
DOI: https://doi.org/10.7554/eLife.47098.005

**Figure supplement 1.** FET family IDRs are not recruited to RCs.
DOI: https://doi.org/10.7554/eLife.47098.003

**Figure 1—video 1.** Time lapse movie of HaloTag-Pol II after HSV1 infection.
DOI: https://doi.org/10.7554/eLife.47098.006

**Figure 1—video 2.** Time lapse movie of HaloTag-Pol II after HSV1 infection.
DOI: https://doi.org/10.7554/eLife.47098.007

vitro, the IE and tegument proteins are even slightly more disordered (*Figure 1E*; *Figure 1—source data 1*). Under the working hypothesis that interactions between IDRs drive phase separation, the similarity in predicted disorder profiles between our curated list and the IE and tegument proteins suggests that IDRs in viral proteins may be as likely to undergo LLPS as experimentally validated proteins.

Based on the above descriptive observations, we hypothesized that Pol II should be recruited to RCs through interactions between its CTD and other IDR-containing proteins within the RC. To test this, we measured the Fluorescence Recovery After Photobleaching (FRAP) dynamics of Pol II in RCs. We saw a consistent slowing of recovery as infection progressed (*Figure 1F*), which could be interpreted as evidence that Pol II is incorporated and sequestered within the RC, an 'ageing' phenotype that others have described (*Shin et al., 2017*). Subsequent experiments to directly test this hypothesis, however, cast doubt on this interpretation.

Hub formation by Pol II in uninfected cells occurs in a manner dependent on the length of the Pol II CTD, a prominent IDR (*Boehning et al., 2018*). To test whether the Pol II CTD likewise mediates interaction with RCs, we compared Pol II accumulation in RCs using the cells generated by Boehning and colleagues: wild-type Pol II CTD (with 52 heptad repeats), and with truncated (25 repeats) or extended (70 repeats) CTDs. Despite a strong effect in uninfected cells on the distribution of Pol II (*Boehning et al., 2018*), the length of the CTD had no detectable effect on Pol II incorporation into RCs (*Figure 1G*), suggesting that Pol II recruitment in not sensitive to CTD length.

As a further test of the role of IDR interactions in Pol II accumulation within RCs, we treated cells with 1,6-hexanediol, which disrupts weak hydrophobic interactions between IDRs that drive LLPS (*Lin et al., 2016*). We infected cells for 5 hr, and then subjected them to treatment with a high concentration (10% w/v) of 1,6-hexanediol. Despite significant morphological changes in the nucleus after treatment, consistent with widespread disruption of cellular organization (*Lin et al., 2016*), Pol II remained highly enriched in RCs (*Figure 1H*). Furthermore, other IDRs with LLPS capabilities and which are known to interact with the CTD (*Chong et al., 2018*) are not enriched in RCs (*Figure 1—figure supplement 1*), suggesting that formation of RCs does not require interactions between the IDRs of Pol II and other host or viral proteins.

## Unrestricted Pol II diffusion across RC boundaries is inconsistent with an LLPS model

The data outlined in *Figure 1* present a potential contradiction, as RCs exhibit several properties commonly associated with phase separation in vitro, yet Pol II recruitment to RCs is clearly not dominated by homo- or heterotypic interactions through its IDR. We sought to better understand the mechanism driving the enrichment of Pol II in RCs by measuring the behavior of individual Pol II

molecules. To accurately capture both immobile and freely diffusing Pol II molecules, we used stroboscopic photo-activatable single particle tracking (spaSPT) to visualize and track molecules (*Figure 2A*) (*Hansen et al., 2017*; *Hansen et al., 2018*). We labeled Halo-Pol II with equal amounts of $JF_{549}$ and PA-$JF_{646}$ (*Grimm et al., 2015*; *Grimm et al., 2016*), allowing us to accurately generate masks to then sort trajectories as either 'inside' or 'outside' of RCs (*Figure 2B*, *Figure 2—video 1* and *2*). A qualitative comparison of trajectories of single Pol II molecules in RCs shows enrichment in short, constrained jumps compared to uninfected cells (*Figure 2C*, red arrows).

Quantitative measurements can be made by building histograms of all the displacement distances from the trajectories, and fitting to a two-state model in which Pol II can either be freely diffusing ('free'), or immobile and hence presumably bound to DNA ('bound') (*Figure 2D*, inset). Such a two-state model gives two important pieces of information: the fraction of 'bound' and 'free' molecules, and the apparent diffusion coefficient of each population (*Hansen et al., 2018*). It is important to note that, because this modeling approach takes the aggregate of many thousands of traces, these data cannot measure how long a particular molecule remains bound in a given binding event. Therefore, 'bound' refers to both specific DNA binding events—for example molecules assembled at a promoter or engaged in mRNA elongation—as well as transient, non-specific binding interactions.

The difference in the behavior of Pol II inside RCs compared with the rest of the nucleoplasm is immediately apparent from examining the lengths of jumps between consecutive frames (*Figure 2C, D*). Surprisingly, the mean apparent diffusion coefficient of the free population was unchanged between trajectories inside of RCs compared with those outside RCs or in uninfected cells (*Figure 2E*; *Figure 2—figure supplement 1A–C*). If RCs were a *bona fide* separate phase, one would expect differences in molecular crowding or intermolecular interactions to predominantly affect free diffusion, resulting in substantially different diffusion coefficients.

To confirm this result, we performed a fluorescence loss in photobleaching (FLIP) experiment, in which a strong bleaching laser targets the inside of an RC and loss of fluorescence elsewhere in the nucleus is measured to quantify exchange of Pol II between the nucleoplasm and the RC. Consistent with the spaSPT data, we see that Pol II molecules exchange between RCs and the rest of the nucleoplasm as fast as Pol II in uninfected cells (*Figure 2F*). Similar results were obtained by using Pol II tagged with the photo-convertible fluorescent protein Dendra2 (*Cisse et al., 2013*) and photo-converting, rather than bleaching, molecules in the RC (*Figure 2—figure supplement 2A*). Unlike the FRAP data, the rate of photobleaching does not change as a function of time after infection (*Figure 2—figure supplement 2B–C*). Thus, Pol II molecules freely diffuse out of the RC, rather than remain sequestered within RCs.

An LLPS model predicts that a diffusing Pol II molecule within an RC should be more likely to remain within the RC than to exit when it reaches the compartment boundary. We tested this prediction by examining all trajectories for events in which a molecule crosses from inside RCs to outside, or vice versa, to look for evidence of such a boundary constraint. Comparing the distribution of displacements for a particle going from inside the RC to outside, we see no difference in the distribution of displacements, either entering or leaving RCs, when compared to uninfected cells in which mock RC annotations were randomly imposed in silico (*Figure 2G*; *Figure 2—figure supplement 3*). Indeed, we cannot detect any evidence of a boundary for molecules entering or leaving RCs, further arguing that RCs do not consist of a distinct liquid phase.

While the two-state model shows no change in diffusion coefficient of Pol II, the fraction of molecules in the 'bound' state doubles inside RCs, reaching ~70% (*Figure 2H*). We verified that this was not an artifact of the masking process by randomly shuffling RC annotations around in silico (*Figure 2—figure supplement 3C,D*), and that diffusion coefficients of the bound population are consistent with those of chromatin (*Hansen et al., 2018*), and thus reflect DNA binding (*Figure 2—figure supplement 1D*). The increase in the fraction of bound molecules is further supported by slowed recovery in the FRAP data (*Figure 1F*). The striking shift in the fraction of DNA-bound molecules, even while the FLIP decay rates remain unchanged, argues that this is due to an increase in the rate of Pol II binding rather than a decrease in the rate of Pol II unbinding. Thus, the mechanism driving Pol II recruitment to RCs is dominated by DNA binding rather than unbinding, which argues against the 'aging' phenomenon that others have observed (*Shin et al., 2017*).

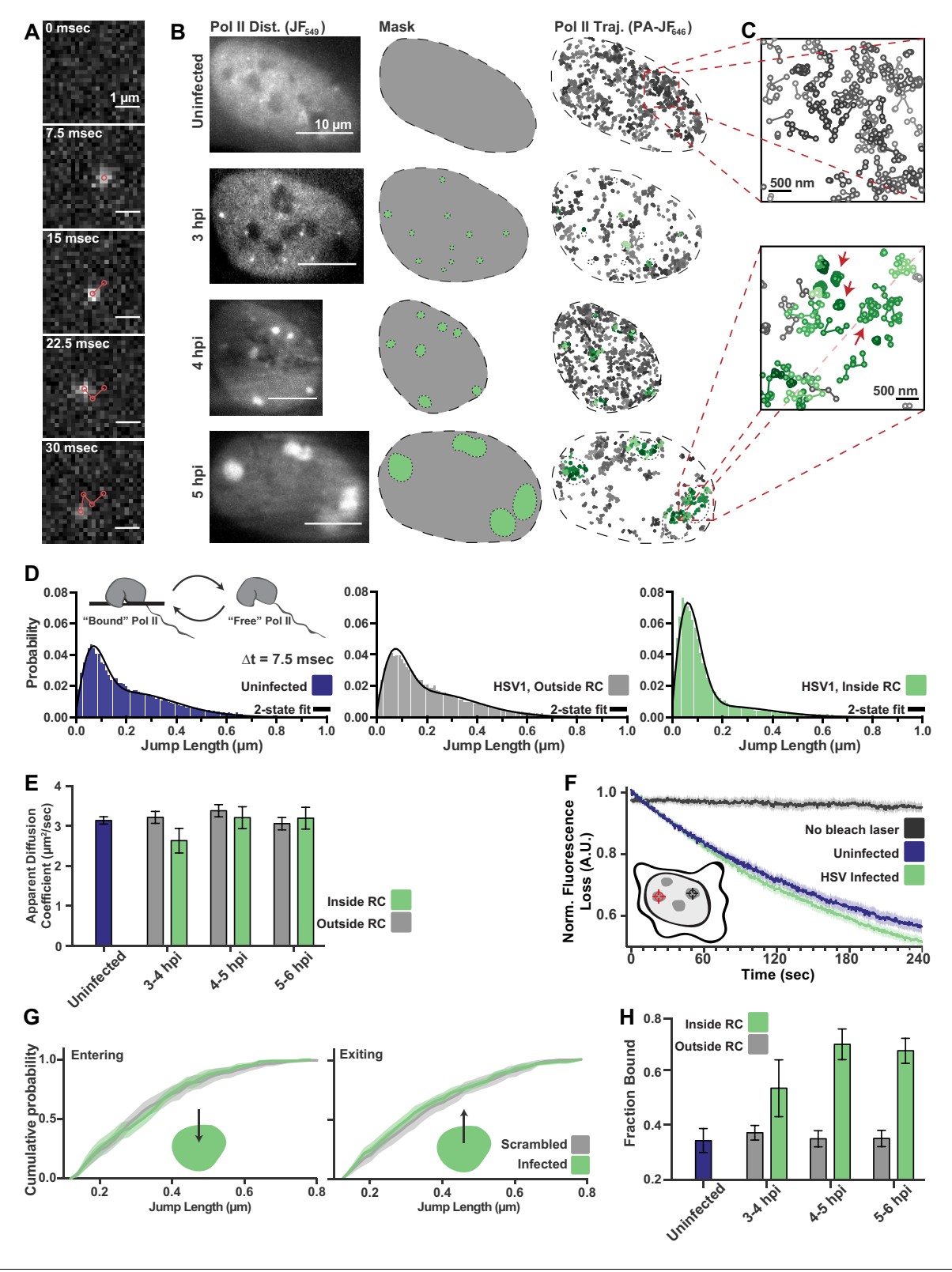

**Figure 2.** spaSPT of Pol II in infected cells shows no change in diffusion but an increase in binding. (**A**) Example frames from spaSTP localization and tracking. Scale bar is 1 μm. (**B**) spaSPT experiments in infected cells at different times post infection. RCs are identified using Pol II fluorescence and used to make masks for sorting trajectories (green inside RCs; gray outside). (**C**) Zoom-in of trajectories in infected and uninfected cells. Red arrows show examples of traces with restricted movement. (**D**) Jump length distributions between consecutive frames of spaSPT trajectories. Histograms

*Figure 2 continued on next page*

*Figure 2 continued*

pooled from uninfected cells (n = 27), or HSV1 infected cells between 4 and 6 hpi (n = 96). Each distribution is fit with a two-state model. Inset shows depiction of two-state model where Pol II can either be freely diffusing or DNA-bound. (E) Mean apparent diffusion coefficient from the two-state fit in (D). Error bars are the standard deviation of the mean, calculated as described in Materials and methods. (F) FLIP curves comparing the rate of fluorescence loss after photobleaching Pol II in uninfected and HSV1 infected cells. Schematic shows location of bleaching laser (red crosshairs) and the region measured (black crosshairs). (G) Cumulative distribution function of the mean flanked by the SEM for jump lengths of molecules entering (left) or exiting (right) RCs. The distribution for HSV1-infected cells is compared to the distribution of jump lengths when RC annotations have been shuffled randomly. (H) Mean fraction of bound molecules from the two-state fit in (D). Error bars are the standard deviation of the mean, calculated as described in Materials and methods.

DOI: https://doi.org/10.7554/eLife.47098.008

The following video and figure supplements are available for figure 2:

**Figure supplement 1.** Sampling statistics and quality measurements of spaSPT.
DOI: https://doi.org/10.7554/eLife.47098.009
**Figure supplement 2.** FLIP shows exchange within and between RCs.
DOI: https://doi.org/10.7554/eLife.47098.010
**Figure supplement 3.** Comparison of *bona fide* RCs with RCs generated in silico.
DOI: https://doi.org/10.7554/eLife.47098.011
**Figure 2—video 1.** Example of SPT data from an uninfected cell.
DOI: https://doi.org/10.7554/eLife.47098.012
**Figure 2—video 2.** Example of SPT data from a cell 4 hpi.
DOI: https://doi.org/10.7554/eLife.47098.013

## Pol II recruitment to RCs occurs independent of transcription initiation

One possible explanation for the increased fraction of bound Pol II in RCs would be a high level of active transcription in these compartments. Multiple lines of evidence suggest that transcription derived from the viral genome is activated to a much greater extent than transcription of even the most highly transcribed host mRNAs (*Rutkowski et al., 2015*), and this may be sufficient to explain the increase in DNA-bound Pol II.

To test whether active transcription is driving Pol II recruitment to RCs, we treated infected cells with either Triptolide or Flavopiridol, small molecules that selectively inhibit stable Pol II promoter binding or transcription initiation, respectively (*Figure 3A*) (*Bensaude, 2011*). HSV1 requires the expression of immediate-early and early genes to generate its DNA replication machinery, so we allowed the infection to progress for four hours before treating with either compound. Cells at this time point have well-formed RCs, and Pol II binding is already greatly increased (*Figure 2H*). We treated these cells with either drug for 15, 30, or 45 min to inhibit de novo transcription and allow any elongating polymerases to finish transcribing (*Figure 3B*). RNA fluorescence in situ hybridization (FISH) against an intronic region showed significantly reduced nascent transcripts after 30 min of drug treatment (*Figure 3C,D*). Remarkably, even after 45 min of treatment, ~80% of the Pol II signal remains within RCs (*Figure 3E,F*). These data suggest that the recruitment of Pol II to RCs occurs largely independently of transcription, and without stable engagement with gene promoters.

By spaSPT, in uninfected cells, Triptolide or Flavopiridol treatment both reduce the fraction of bound Pol II by half, to ~15% (*Figure 3G*), similar to what others have reported (*Boehning et al., 2018*; *Teves et al., 2018*). Nevertheless, inhibition of transcription with Flavopiridol reduced the bound fraction inside of RCs by only ~5% (*Figure 3G*). Even treatment with Triptolide, which prevents stable engagement with TSS-proximal DNA, only reduced the fraction bound by ~12% (*Figure 3G*). Given this result, we conclude that the majority of binding events we measure are independent of viral transcription.

HSV1 infection appears also to confer some resistance to the effects of these drugs on Pol II binding to host chromatin, despite the fact that these inhibitors are sufficient to abrogate transcription (*Figure 3C–F*). Given the inherent limitation of spaSPT for inferring the length of binding events, we wanted to confirm that drug treatment prevented stable Pol II binding. Indeed, FRAP experiments in cells treated with Triptolide show a dramatically faster recovery rate for both uninfected and infected cells (*Figure 3H*). For the infected samples, this means that the 'bound' molecules measured by SPT do not remain bound for long times, as one would expect from high affinity protein-protein or protein-DNA interactions at cognate sites. Instead, the majority of the bound fraction is comprised of

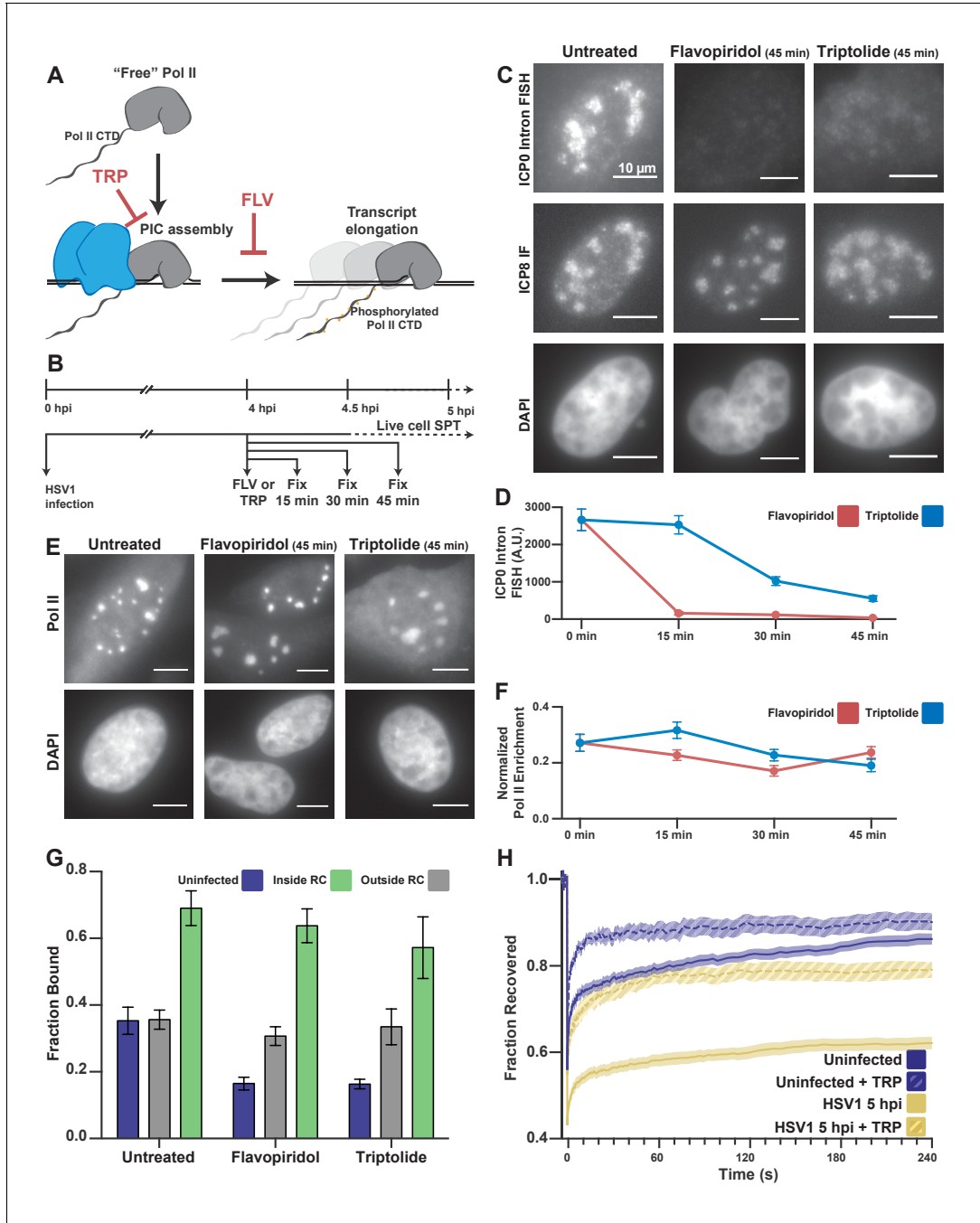

**Figure 3.** Pol II recruitment to RCs occurs independent of active transcription. (**A**) Schematic of Pol II-mediated transcription inhibition. (**B**) Schematic of the experiment regimen for imaging infected cells after transcription inhibition. (**C**) RNA FISH against the ICP0 intron to measure nascent transcription after Flavopiridol or Triptolide treatment. ICP8 marks viral RCs. (**D**) Quantification of the ICP0 intron signal in untreated cells (n = 170 RCs) those treated with TRP(n = 192, 171, 191 RCs, respectively) and FLV(n = 158, 238, 153 RCs, respectively). Error bars are standard error of the mean. (**E**) Halo-Pol II distribution after 45 min of Triptolide or Flavopiridol treatment. All scale bars are 10 μm. **F**) Quantification of the total fraction of Pol II recruited to RCs in untreated cells (n = 29) with TRP(n = 33, 24, 33, respectively) and FLV(n = 36, 24, 38, respectively). Error bars represent standard error of the mean. (**G**) Mean fraction bound measured from spaSPT of Halo-Pol II, after transcription inhibition. Error bars are the standard deviation of the mean, calculated as described in STAR methods. (**H**) FRAP recovery curves of Pol II with (hashed) and without (solid) Triptolide treatment, for uninfected cells (n = 31, nine respectively) and cells infected with HSV1, 5hpi (n = 32, 12 respectively).

DOI: https://doi.org/10.7554/eLife.47098.014

The following figure supplement is available for figure 3:

**Figure supplement 1.** HSV1 mutants affect neither Pol II recruitment nor binding dynamics.

*Figure 3 continued on next page*

*Figure 3 continued*

DOI: https://doi.org/10.7554/eLife.47098.015

transient binding events independent of transcription. The fact that infected cells show increased DNA binding outside of RCs after drug treatment may be a result of other viral mechanisms that occur during infection, such as aberrant Pol II CTD phosphorylation (*Rice et al., 1994*) or termination defects (*Rutkowski et al., 2015*). Still, our results suggest that viral DNA and/or DNA-associated proteins mediate rapid and predominantly nonspecific interactions with Pol II in RCs.

It has been reported that the viral protein ICP8 interacts with the CTD of Pol II through a bridging interaction by the viral protein ICP27 (*Zhou and Knipe, 2002*). Others have used ICP27 truncation mutants to suggest that this ICP27-mediated mechanism is responsible for Pol II recruitment into RCs (*Dai-Ju et al., 2006*). Thus, we tested HSV1 mutant strains n504 and n406, which carry nonsense mutations in ICP27 that weaken or abrogate (respectively) the Pol II-ICP8 interaction, and should be defective for Pol II recruitment to RCs (*Rice and Knipe, 1990*; *Zhou and Knipe, 2002*). While these mutant strains generally show a deficiency in forming RCs and producing virus, we found that in cells where RCs do form, Pol II is recruited as efficiently as in cells infected with a WT virus (*Figure 3—figure supplement 1A*), and the FRAP recovery dynamics are indistinguishable from WT virus-infected cells (*Figure 3—figure supplement 1B*) suggesting it is unlikely that this specific viral complex is the major player in recruiting Pol II to RCs.

## HSV1 DNA is much more accessible than host chromatin to Pol II

The finding that Pol II molecules remain bound—however transiently—to the viral DNA, even in the absence of transcription or other interactions involving viral proteins, suggests that the DNA itself could plays a dominant role in Pol II enrichment in RCs. Knowing the amount of viral DNA contained in any one RC may be crucial to understand the role viral DNA may play in RC formation and function, but to our knowledge, this has not been determined. We therefore sought to measure the amount of DNA in RCs using DNA FISH by targeting fluorescent probes to two specific regions of the viral genome (*Figure 4A*). Fluorescence intensities from infected samples were compared at different times post infection to samples that were infected in the presence of phosphonoacetic acid (PAA), an inhibitor of viral DNA replication that ensures there is only one copy of the viral genome per punctum (*Figure 4B*; *Figure 4—figure supplement 1A*) (*Eriksson and Schinazi, 1989*).

The number of genomes within an RC correlates well with the time post infection (*Figure 4C*), and there is also a strong correlation between RC size and genome copy number (*Figure 4—figure supplement 1B*). Based on these data, we calculate that the average RC at 6 hpi has a DNA concentration of $3.9 \times 10^4$ bp/$\mu$m$^3$, approximately 240 times less concentrated than average host chromatin (*Monier et al., 2000*). The totality of viral DNA in an average cell after 6 hr of infection corresponds to just ~0.2% of total DNA in karyotypically normal human nuclei (*Table 1*). Yet, despite its 100-fold lower DNA concentration, inhibition of viral DNA replication with PAA caused the fraction of bound Pol II molecules inside the pre-replication foci to decrease to ~50% (*Figure 4D*).

Since most of the observed Pol II binding events that we observe inside of RCs appear to be unrelated to transcription, but are clearly dependent on viral DNA replication, we wondered what might be different about the viral genome relative to host chromosomes. A likely candidate is the chromatin state of the viral DNA. There is presently no consensus about the organization of viral DNA during lytic infection, but mass spectrometry studies have failed to detect histones associated with viral DNA (*Dembowski and DeLuca, 2015*). Moreover, infection of a cell line constitutively expressing Histone H2B fused to HaloTag is not incorporated into RCs (*Figure 4E*).

To measure histone occupancy on HSV1 DNA, and get a measure of its accessibility, we turned to ATAC-seq, which gives signal proportional to the accessibility of the DNA at a given locus (*Buenrostro et al., 2013*). Based on the amount of viral DNA present in an infected cell, we calculated the fraction of reads one would expect to map to the virus relative to the host. At 6 hpi, by DNA FISH the viral DNA represents an average 0.2% of total nuclear DNA content. Yet under the same conditions at this time point, 24.2% of reads mapped to the virus on average, showing that viral DNA is at least 100-fold more accessible (*Table 1*).

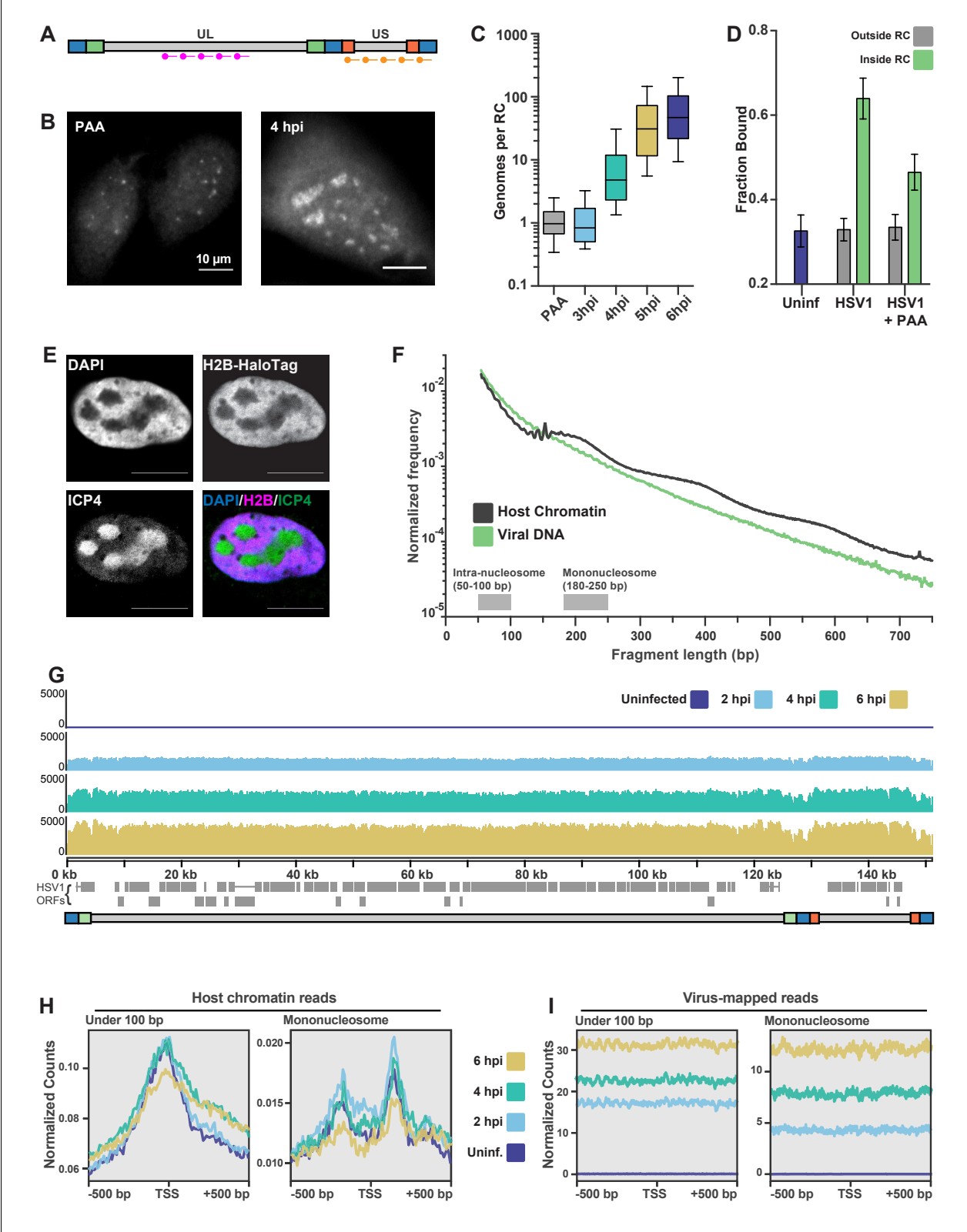

**Figure 4.** ATAC-seq reveals HSV1 DNA is much more accessible than chromatin. (**A**) Schematic of the Oligopaint targets for DNA FISH. Separate probe sets target regions in the Unique Long (UL) arm and the Unique Short (US) arm. **B**) Representative images of DNA FISH of cells four hpi, infected in the presence (PAA, left) or absence (4hpi, right) of the replication inhibitor PAA. Pixel intensity values are the same for the two images. Scale bars are 10 μm. (**C**) Fluorescence intensity of DNA FISH signal in RCs after infection. 5–95% intervals are shown, with inner quartiles and median. Data are

*Figure 4 continued on next page*

*Figure 4 continued*

normalized to the median intensity value of PAA-treated infected cells. Medians: PAA = 1.0, 3 hpi = 0.8, 4 hpi = 4.8, 5 hpi = 31.1, 6 hpi = 47.0. (D) Mean fraction bound for Pol II in infected cells with and without PAA. Error bars are the standard deviation of the mean, calculated as described in Materials and methods. (E) H2B-Halo cells show histone H2B is not incorporated into RCs. Innumofluorescence against ICP4 marks RCs. (F) Fragment length distribution of ATAC-seq data for cells 4 hpi. Lengths corresponding to intra-nucleosomal DNA (50–100 bp) and mononucleosomal DNA (180–250 bp) are marked as a reference. (G) ATAC-seq read density plotted across HSV1 genomic coordinates. (H) ATAC-seq analysis of intra-nucleosomal DNA (50–100 bp) and mononucleosomal DNA (180–250 bp). Global analysis of all human Pol II-transcribed genes, centered at the transcription start site (TSS). (I) The same analysis as in (G), but centered at the TSS of HSV1 genes.

DOI: https://doi.org/10.7554/eLife.47098.016

The following figure supplement is available for figure 4:

**Figure supplement 1.** Quantification of DNA content and chromatin state in HSV1 RCs.

DOI: https://doi.org/10.7554/eLife.47098.017

The ATAC-seq fragment length distributions (*Figure 4F*; *Figure 4—figure supplement 1C*) showed a much faster decay for reads mapping to the virus at all times post infection, and with no evidence of nucleosomal laddering, in stark contrast to reads that map to the host genome. When we visualized the position of all HSV1-mapped reads along the viral genome, the profiles were strikingly flat and featureless (*Figure 4G*). An average of all annotated human mRNA genes, centered at the TSS, shows a characteristic peak of accessibility at the TSS for reads with a length corresponding to inter-nucleosomal distances (<100 bp), and a characteristic trough of mono-nucleosome sized fragments (180–250 bp) (*Figure 4H*). By contrast, TSS averages mapped to the viral genome for either short or mono-nucleosome fragments show no changes in accessibility. Even averaging over all viral transcripts, it is clear that the entire viral DNA remains equally accessible (*Figure 4I*). Taken together, these data indicate that the HSV genome is maintained in a largely nucleosome-free state, and thus highly accessible to DNA binding proteins like Pol II.

## Transient DNA-protein interactions drive Pol II hub formation through repetitive exploration of the replication compartment

Knowing that the DNA inside RCs is vastly more accessible to nuclear factors than host chromatin, we next asked what emergent properties of this accessible DNA might help explain Pol II recruitment. Using an HSV1 strain that allows incorporation of nucleotide analogs, (*Dembowski and DeLuca, 2015*), we fluorescently labeled DNA, imaged it at super-resolution, and found that, within a given RC, viral DNA shows variability in local density of nearly three orders of magnitude (*Figure 5A*).

The greater accessibility and higher variability in local density of viral DNA lend themselves to a possible mechanism by which Pol II becomes enriched. Recent theoretical work has shown that a

**Table 1.** Quantitative measurements of HSV1 DNA inside of RCs.

Related to *Figure 4*. Using the values obtained through DNA FISH and ATAC-seq, we can make estimates of the copy number, concentrations, and relative enrichment of the viral DNA compared to the host. All values are calculated based on measurements of cells 6 hpi.

| Table 1 | Genome Size (bp) | Genome Copy number[‡] | Total DNA (bp) | Percent of Total DNA[‡] | Concentration (bp/µm$^3$)[§] | ATAC-seq read percentage[¶] | Fold enrichment over expected[**] |
|---|---|---|---|---|---|---|---|
| Host Genome[*] | $3.2 \times 10^9$ | 2 | $6.4 \times 10^9$ | 99.8 (±0.2) | 9.4 (±1.6) x10$^6$ | 75.8 (±10.4) | 0.8 (±0.1) |
| Viral DNA | $1.5 \times 10^5$ | 82 (±105) | 1.3 (±1.6) x10$^7$ | 0.2 (±0.2) | 3.9 (±5.8) x10$^4$ | 24.2 (±10.4) | 130 (±170) |
| Rel. Diff.[†] | $2.1 \times 10^4$ | | 513 (±658) | | 240 (±369) | | |

All values are the Mean (±S.D.).

*. Assuming karyotypically normal human cell; †. relative difference = Human/HSV1; ‡. Under experimental conditions of MOI = 1; §. Concentration assuming nucleus volume taken from *Monier et al. (2000)*; ¶. based on total reads mapped from each organism, n = 3; ** Fold enrichment = ATAC seq read percentage/Percent of Total DNA.

DOI: https://doi.org/10.7554/eLife.47098.018

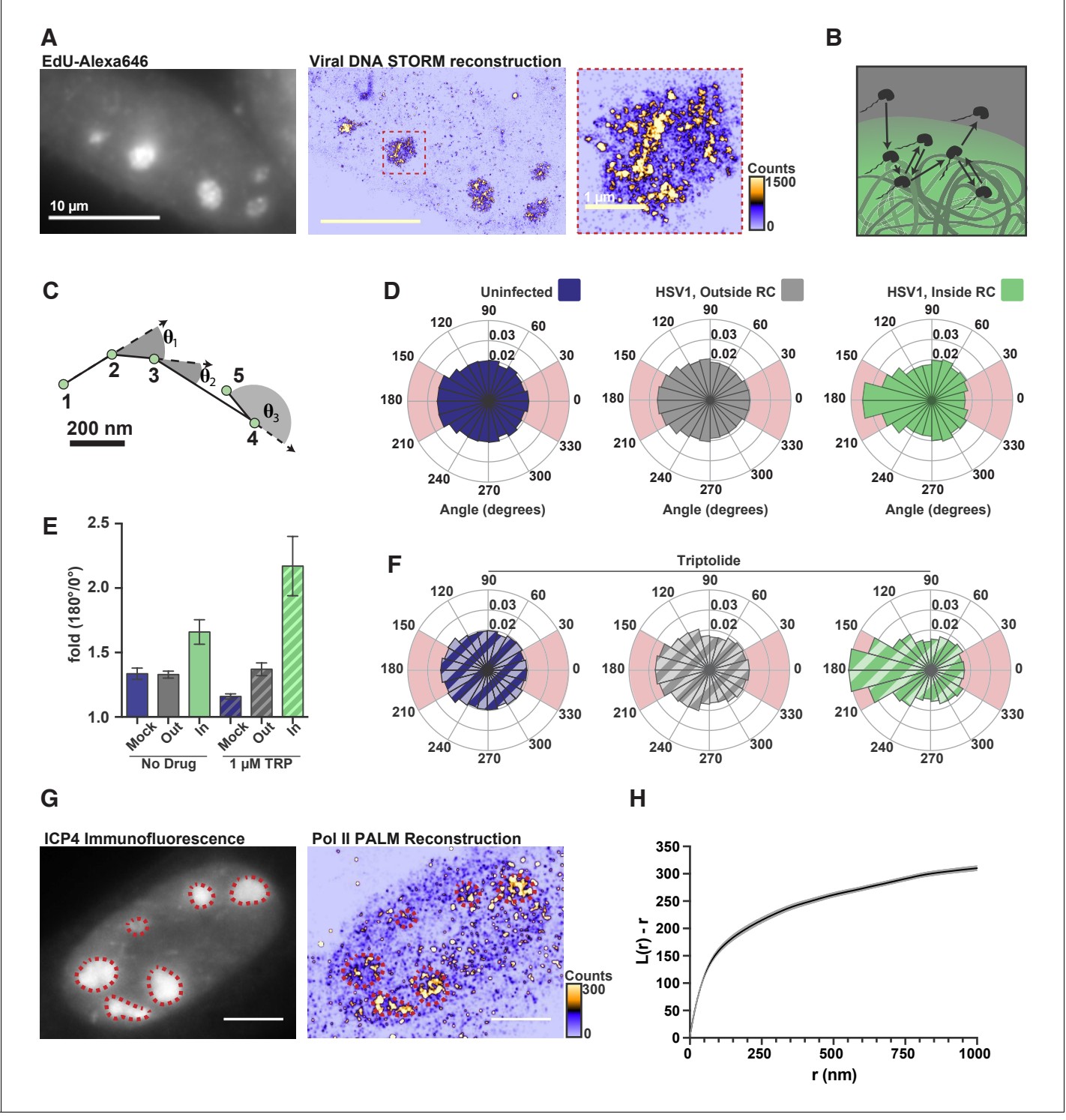

**Figure 5.** DNA-binding alters Pol II exploration of RCs. (**A**) STORM image of fluorescently labeled HSV1 DNA. Zoom-in shows one RC, and the heatmap shows the number of fluorophore localizations in each rendered pixel. (**B**) Schematic of Pol II exploring an RC and randomly sampling the viral DNA. (**C**) Example spaSPT trace, marking the angles between consecutive steps. (**D**) Angular distribution histograms extracted from Halo-Pol II in uninfected cells, and HSV1 infected cells 4–6 hpi, inside and outside of RCs. (**E**) Quantification of the relative probability of moving backward compared to forward (180°±30°/0°±30°). Error bars are the standard deviation of the mean, calculated as described in Materials and methods. (**F**) Same as in (**D**), except that cells were treated with Triptolide at least 30 min prior to imaging. Quantification of this data is also show in (**E**). (**G**) Representative PALM image of Halo-Pol II. ICP4 marks viral RCs. Heatmap corresponds to the number of detections per rendered pixel. (**H**) L-modified Ripley Curve (L(r)-r) for

*Figure 5 continued on next page*

*Figure 5 continued*

Halo-Pol II inside of RCs in cells five hpi (n = 13 cells). Graph shows the mean flanked by the SEM. All scale bars are 10 µm. Also see *Figure 2—figure supplement 3E and F*.

DOI: https://doi.org/10.7554/eLife.47098.019

polymer like DNA, which has many binding sites in close proximity, can induce an interacting protein to revisit the same or adjacent sites repetitively during its exploration of the nucleus (*Amitai, 2018*) (*Figure 5B*). In such a case, we should be able to see signatures in our spaSPT dataset of Pol II continually revisiting adjacent sites on the viral DNA. To check, we calculated the angle formed by consecutive displacements and compiled these angles into a histogram (*Figure 5C*) (*Izeddin et al., 2014*). For particles experiencing ideal Brownian motion, the angular histogram will be isotropic. Anisotropy can arise through a variety of mechanisms, such as adding the aforementioned 'traps,' thereby giving the particle a greater probability of revisiting proximal sites before diffusing away (*Amitai, 2018*).

In uninfected cells, and in infected cells outside of RCs, Pol II displays diffusion that is largely isotropic. In stark contrast, inside RCs Pol II diffusion is highly anisotropic, particularly around 180° (*Figure 5D*; *Figure 2—figure supplement 3E*; *Figure 2—figure supplement 3F*). To compare across samples, we computed the likelihood of a backward translocation (180°±30°) relative to a forward translocation (0°±30°). Analyzed this way, Pol II inside RCs has a 1.7-fold greater chance of making a backward step for every forward step it takes (*Figure 5E*). In cells treated with Triptolide, where stable binding is inhibited, the effect created by transient binding events is further amplified (*Figure 5E,F*), which helps explain the dramatic retention of Pol II inside RCs, even 45 min after inhibition of transcription (*Figure 3E*). These data are most consistent with a model in which Pol II repetitively visits the highly accessible viral genome via multiple weak, transient binding events which likely result in Pol II hopping or sliding along the DNA. The sharp anisotropy of the molecular exploration within the compartment means that a given Pol II molecule within an RC is more likely to visit the same or proximal sites multiple times before either finding a stable binding site or diffusing away.

The heterogeneous distribution of viral DNA within RCs, and the anisotropic way Pol II explores RCs, is also borne out in the distribution of Pol II molecules. Similar to the viral DNA, super-resolution photo-activated localization microscopy (PALM) renderings of infected nuclei revealed a heterogeneous Pol II distribution within RCs (*Figure 5G*). A key prediction of the formation of phase condensates is that LLPS compartments should form at a characteristic critical concentration, and that molecules within the high concentration phase should return to homogeneity within the phase (*Freeman Rosenzweig et al., 2017*). The highly heterogeneous nature of Pol II within the RCs provides yet further evidence that these compartments are not derived through an LLPS process. We used Ripley's L-function to measure how the Pol II distribution deviates from spatial randomness, with values greater than zero indicating a concentration higher than predicted for complete randomness at that given radius (*Figure 5H*) (*Ripley, 1977*). We find that the curve remains well above zero, and increases, for all radii up to one micron. This suggests that Pol II forms hubs within RCs at multiple length scales, consistent with the behavior of Pol II in uninfected cells (*Boehning et al., 2018*), and inconsistent LLPS driving the constitution of RCs.

## Nonspecific interactions with viral DNA license recruitment of other proteins

Seeing that Pol II is recruited to RCs via transient and nonspecific binding to the viral genome made us wonder whether this effect was specific to Pol II, or whether DNA accessibility can generally drive the recruitment of any DNA-binding proteins to RCs. Certainly, many other DNA-binding proteins are recruited to RCs (*Dembowski and DeLuca, 2015*). To assess whether nonspecific DNA binding could be responsible for their accumulation as well, we looked to an extreme example: The tetracycline repressor (TetR), and the Lac repressor (LacI). Both proteins are sequence-specific bacterial transcriptions factor, the consensus sites for which are absent in both human and HSV1 genomes. If proteins like TetR and LacI can be recruited to RCs despite lacking cognate binding sites, this is strong evidence that nonspecific DNA association is the driving mechanism for recruitment.

Expression of TetR-Halo and LacI-Halo shows enrichment within RCs (*Figure 6*), in stark contrast to Halo-NLS or HaloTag-fused IDRs (*Figure 1—figure supplement 1*). Furthermore, a comparison of the jump lengths measured in single particle tracking of TetR-Halo also reveals an enrichment in short translocations inside of RCs, consistent with higher fraction of bound TetR-Halo molecules (*Figure 6—figure supplement 1*). Thus, while IDR-based interactions alone are unable to generate strong enrichment in the RCs (*Figure 1—figure supplement 1*), even modest nonspecific DNA-binding affinity appears sufficient to do so.

These data suggest a model in which viral Pol II recruitment consists of transient, nonspecific binding/scanning events of the highly exposed viral genome (*Figure 7A*). A DNA-binding protein exploring the nucleus (uninfected, or infected but outside of RCs) may encounter some occasions for nonspecific interaction with duplex DNA, but because of the nucleosome-bound nature of the host chromatin, these binding/scanning events are necessarily spatially dispersed and infrequent (*Figure 7B*). Within RCs, many copies of the unprotected HSV1 DNA are present, allowing nonspecific events to happen much more frequently, with fewer and shorter 3D excursions between DNA contacts (*Figure 7C*). Thus, transient protein-DNA interactions drive enrichment of DNA-binding proteins within RCs.

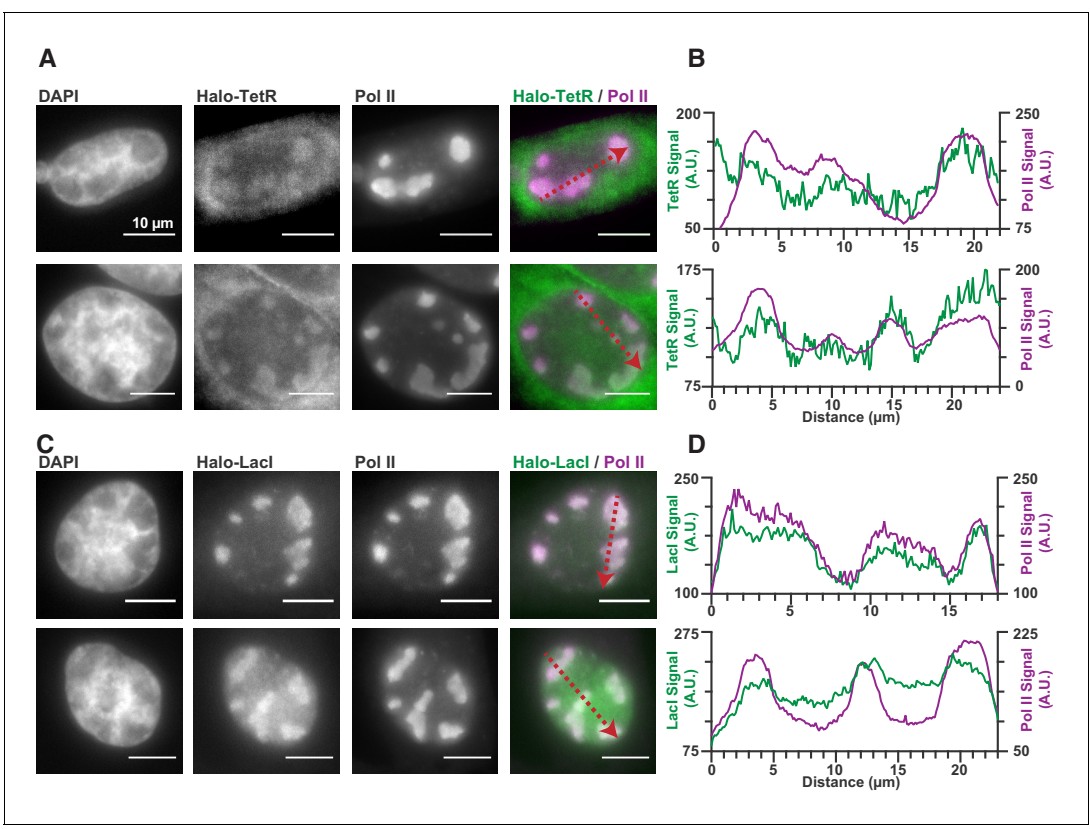

**Figure 6.** Nonspecific DNA binding drives accumulation of other factors in RCs. (**A and C**) Two representative cells from SNAPtag-RPB1 cells expressing TetR-Halo (**A**) and LacI-Halo (**C**), showing that both bacterial transcription factors are enriched in RCs. (**B and D**) Pixel line scans of images in (**A**) and (**C**). Red arrows give the direction of the x-axis. Left y-axis is the intensity of TetR-Halo or LacI-Halo fluorescence, right y-axis is the intensity of SNAPtag–Pol II fluorescence. All scale bars are 10 μm. Also see *Figure 1—figure supplement 1*.

DOI: https://doi.org/10.7554/eLife.47098.020

The following figure supplement is available for figure 6:

**Figure supplement 1.** SPT of Halo-TetR in infected cells.
DOI: https://doi.org/10.7554/eLife.47098.021

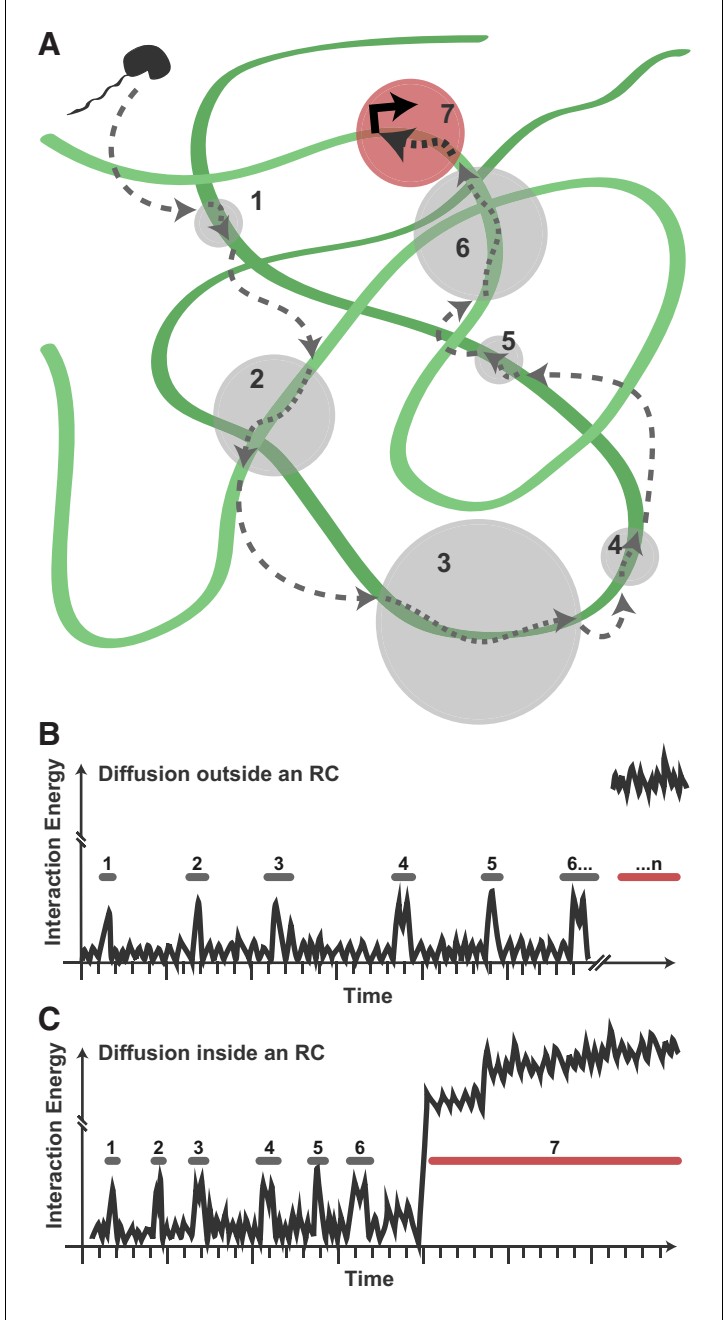

**Figure 7.** Model for Pol II exploration of RCs. (**A**) A Pol II molecule encounters the accessible viral DNA multiple times along one potential route to eventually bind at a promoter. 3D diffusion through the RC is interrupted by binding interactions with the viral DNA (gray circles). (**B**) Hypothetical comparison of nuclear exploration outside RCs as a function of time and binding energy. A DNA-binding protein in the chromatinized nucleus will encounter nucleosome-free DNA sporadically, making multiple low-affinity interactions before eventually finding a high-affinity site. (**C**) Inside an RC, the high DNA accessibility might shorten the length of 3D excursions before a DNA-binding protein encounters another region of viral DNA in a low-affinity, nonspecific interaction. This, in turn, may reduce the distance a molecule might diffuse before its next binding event, and increases both the chances of that molecule remaining in close proximity and the chances that it will find a high binding energy interaction.
DOI: https://doi.org/10.7554/eLife.47098.022

## Discussion

### Multiple routes to create high local concentrations

Here, we have demonstrated that Herpes Simplex Virus type one accumulates Pol II in replication compartments because the virus' unusually accessible DNA genome provides many potential non-specific binding sites, acting as a molecular sink which causes a net accumulation of Pol II even in the absence of transcription. Such a mechanism for locally concentrating proteins is revealing, as it neither requires the formation of stable macromolecular structures nor produces any behaviors at the single-molecule level suggesting a separate liquid phase. Instead, by virtue of the fact that the viral genome appear to act as a single polymer globule (*Figure 5A*), from the macroscopic view Pol II recruitment to RCs appears to share many of the behaviors commonly attributed to liquid-liquid phase separation, and yet RCs are clearly a distinct class of membraneless compartment that operate on principles very different from an LLPS model.

We cannot completely rule out the possibility that some form of LLPS-like mechanism contributes to our observations in *Figure 1*. However, our data demonstrate that even if this is the case, it does not contribute to the enrichment of Pol II or the other proteins that we have tested. It is also difficult to rationalize how RCs could exist as a phase condensate without having any measurable impact on the free diffusion (*Figure 2E*), distribution (*Figure 5G,H*) or exchange of molecules that diffuse within and between compartments (*Figure 2F,G*; *Figure 6—figure supplement 1*). Our results prompt the need for a better characterization of *bona fide* phase separation, with a focus on its functional consequences in vivo, and suggest that caution should be exercised before assigning LLPS as the primary assembly mechanism based on criteria such as those applied in *Figure 1*. Likewise, significant caution should be exercised before interpreting the functional role of an LLPS-like system solely based on macroscopic behaviors.

We recently showed that the CTD of Pol II and other Pol II interacting partners can undergo LLPS in vitro and can form hubs in vivo (*Boehning et al., 2018*; *Lu et al., 2018*). Given the data presented above, there appears a contradiction between this and our previous findings. We emphasize that our current results do not mean that interactions between IDRs are not important. Rather, our results suggest an 'upper limit' for the potency Pol II CTD-mediated interactions to facilitate recruitment to RCs. While ectopic over-expression or in vitro preparations of IDRs may spontaneously create droplet-like structures (*Figure 1—figure supplement 1E*), these condensates do not become enriched in RCs either through heterotypic interactions with the Pol II CTD, or with other viral IDRs.

Multiple viral proteins are known to interact with Pol II or other preinitiation complex components. While we tested the most prominent of these interactions, and found that Pol II remains recruited to the viral DNA in the absence of interactions with the viral protein ICP27 (*Figure 3—figure supplement 1*), we cannot—nor do we wish to—rule out the possibility that other viral proteins may help facilitate this process. Importantly, our results do not contradict any of these unique mechanisms, but rather they provide a unifying rationalization for how they may work. As we demonstrated in *Figure 6*, even proteins that would never have been exposed to HSV1 over evolutionary time can still be recruited to RCs, provided they have some nonspecific affinity for DNA. In this way, any protein complex, be it solely viral or host or a composition of both, should be recruited to RCs provided it contains a DNA-binding domain.

### Nonspecific DNA binding is an important feature for nuclear exploration

Our data also reveal a previously underappreciated aspect of how a DNA-binding protein finds its target site within the nucleus. It has long been recognized that nonspecific binding to DNA could accelerate the target search process by sliding in 1D; reducing the search space and empowering faster-than-diffusion association kinetics (*Berg et al., 1981*). The data we present here offer a new perspective on the importance of nonspecific low-affinity binding. When HSV1 replicates its genome, the newly synthesized viral DNA representing just 0.2% of the host chromosome load, is nevertheless, much more accessible to DNA-binding proteins than the totality of host chromatin (*Table 1*).

The finding that Pol II recruitment to RCs is independent of its CTD is reminiscent of RNA Polymerase I (Pol I) transcription of rDNA in the nucleolus. Pol I, lacking the long unstructured CTD that its homolog Pol II contains, is nevertheless robustly recruited to the nucleolus and transcribes rDNA

into ribosomal precursors at prodigious rates. While there are certainly differences in the structure and stability of nucleoli and RCs, it has been shown that nucleolar components indeed exchange with the rest of the nucleoplasm rapidly (*Chen and Huang, 2001*). It is tempting to speculate that recruitment of some nucleolar proteins may benefit from the same mechanism of non-specific DNA binding that drives recruitment of Pol II and other DNA-binding proteins to viral RCs. We speculate that nonspecific protein:nucleic-acid interactions could also be a general mechanism used in other contexts. In particular, many RNA-binding proteins have been reported to undergo apparent LLPS (*Courchaine et al., 2016*), and it will be interesting to explore if these RNA-binding proteins share a similarities to what we observe here.

## Mechanism of Pol II recruitment may explain robust transcription of late genes

An unresolved question in the study of herpesviruses is how genes with seemingly weak promoter elements can sustain such robust transcription (*Rutkowski et al., 2015*). While it is clear that other regulatory components also play a role in regulating late gene transcription (*Davis et al., 2015*), our data may at least help shed light on how the virus robustly transcribes these late genes. After replication onset, when there are many copies of the viral genome present in a single RC, the compartmentalization of Pol II (and the other general transcription factors) mediated through nonspecific binding could greatly favor assembly of PICs at otherwise weak late gene promoters. In this way, the virus can conserve precious sequence space in its genome to encode other important features, relying on fundamental mechanisms of nuclear exploration for Pol II and other components of the transcription machinery while still providing sufficiently robust gene expression for these essential late genes.

## Revisiting insights into chromatin function

DNA accessibility in eukaryotes has long been recognized as a critical parameter for gene regulation (*Paranjape et al., 1994*; *Weintraub and Groudine, 1976*), and many chromatin remodelers have been shown to play a role in modulating nucleosome occupancy at promoters and enhancers. In vivo experiments using sequence-specific eukaryotic transcription factors find that a given factor will spend approximately half its search time undergoing 3D diffusion, and the other half bound nonspecifically, presumably scanning in 1D (*Normanno et al., 2015*); that it may visit as many as $10^5$ noncognate sites during its search. These experiments highlight the challenge a cell faces ensuring that endogenous regulatory sequences are able to effectively compete for cognate DNA-binding factors without becoming adversely influenced by non-target DNA sites. In this context, our results suggest that a less obvious—but critical—function of nucleosomes may involve the passivation of genomic DNA to minimize nonspecific interactions so as to maintain an active pool of freely diffusing nuclear factors, less hindered by their intrinsic propensity for nonspecific binding.

We postulate that a fine balance between the total amount of DNA-binding proteins and the degree of accessible DNA content in the cell is critically important. Nucleosomes, in addition to their obvious structural role in DNA compaction and cis-repression, could serve to uncouple cellular DNA content from the expression level of binding proteins. This mechanism of DNA passivation may be necessary in eukaryotes where the gene density and coding capacity is sparse, but total genomic load is very high; an essential step enabling the evolution of large genomes concomitant with the appearance of chromatin.

This may also point to a less obvious function for the observed increase in accessibility around promoters and enhancers, as a mechanism for effectively funneling DNA-binding proteins into the correct sites. The data presented above suggest that maintaining enhancers and promoters depleted of nucleosomes and accessible to DNA-binding proteins may contribute critically to facilitating the local accumulation of Pol II and other PIC components for transcription activation, without the need to invoke LLPS. In the case of RCs and the recruitment of Poll II, even well-established interactions between IDRs seem to be dispensable, underscoring the diversity of mechanisms driving local hub formation and functional compartments.

# Materials and methods

## Key resources table

| Reagent type (species) or resource | Designation | Source or reference | Identifiers | Additional information |
|---|---|---|---|---|
| Cell line (*Homo sapiens*) | Halo-TAF15 | This paper | U2OS SNAPtag-RPB1, HaloTag-TAF15 | U2OS (15 y/o female osteosarcoma, RRID: CVCL_0042) expressing HaloTag-RPB1(N792D) selected for using alpha-amanitin, further expressing HaloTag-TAF15 (AA 2–205)-NLS and selected for with Hygromycin |
| Cell line (*Homo sapiens*) | H2B-SNAP-Halo | *Hansen et al., 2018* | U2OS Histone H2B-SNA Ptag-HaloTag | U2OS (15 y/o female osteosarcoma, RRID: CVCL_0042) expressing Histone H2B-SNAPtag-HaloTag and maintained in selection with G418 |
| Cell line (*Cercopithecus aethiops*) | Vero | ATCC | ATCC CCL-81; RRID:CVCL_0059 | |
| Cell line (*Cercopithecus aethiops*) | V27 | *Rice and Knipe, 1990* | V27 | Vero cells stable expressing ICP27 under selection of G418. A generous gift from Septhen Rice. |
| Sequence-based reagent | Common DNA FISH forward primer: 5'-GACACGTGATCCGCGATACGAT GAAAGCGCGACGTCAGGTCGGCC-3' | Integrated DNA Technologies | N/A | |
| Sequence-based reagent | Common DNA FISH forward primer: 5'-GACACGTGATCCGCGATACGAT GAAAGCGCGACGTCAGGTCGGCC-3' | Integrated DNA Technologies | N/A | |
| Sequence-based reagent | Common DNA FISH reverse primer: 5'-CTCGCTAATACGACTCACT ATAGCCGGCTCCAGCGG −3' | Integrated DNA Technologies | N/A | |
| Sequence-based reagent | Alexa Fluor 647-labeled RT primer: 5'-TCGCGCTTTCATCGTA TCGCGGATCACGTGTC-Alexa647-3' | Integrated DNA Technologies | N/A | |
| Sequence-based reagent | Alexa Fluor 555-labeled RT primer: 5'-TCGCGCTTTCATCGTAT CGCGGATCACGTGTC-Alexa555-3' | Integrated DNA Technologies | N/A | |

*Continued on next page*

*Continued*

| Reagent type (species) or resource | Designation | Source or reference | Identifiers | Additional information |
|---|---|---|---|---|
| Recombinant DNA reagent | pSNAP-RPB1 (N792D) (plasmid) | This paper | | RPB1 carrying N792D mutation for alpha-amanitin resistence inserted downstream of SNAPtag with the TEV protease sequence as a linker reagion. |
| Recombinant DNA reagent | pHalo-TetR (plasmid) | This paper | | The Tet repressor inserted downstream of HaloTag with the TEV proease site as a short linker. |
| Recombinant DNA reagent | pHalo-LacI (plasmid) | This paper | | The Lac repressor inserted downstream of HaloTag with the TEV proease site as a short linker and a single SV40 NLS at the c-terminus. |
| Recombinant DNA reagent | pHaloTag-3xNLS (plasmid) | *Hansen et al., 2017* | | |
| Recombinant DNA reagent | pHalo-TEV-EWS LC-NLS (plasmid) | *Chong et al., 2018* | | |
| Recombinant DNA reagent | pHalo-TEV-FUS LC-NLS (plasmid) | *Chong et al., 2018* | | |
| Recombinant DNA reagent | pHalo-TEV-Taf15 LC-NLS (plasmid) | *Chong et al., 2018* | | |
| Software, algorithm | Custom implementation of Spot-On and graphical analysis | *Hansen et al., 2018*; this paper | Spot-On | The source code is freely available at https://gitlab.com/dmcswiggen/mcswiggen_et_al_2019 |
| Software, algorithm | Matlab versions 2014b, 2017a | Mathworks | 2014b, 2017a | |
| Software, algorithm | IUPred 2A | *Dosztányi et al., 2005a*; *Dosztányi et al., 2005b* | IUPred | This tool is available at: https://iupred2a.elte.hu/download |
| Software, algorithm | Bowtie2 | *Langmead and Salzberg, 2012* | Bowtie | This tool is availabe at: http://bowtie-bio.sourceforge.net/bowtie2/index.shtml |
| Software, algorithm | SamTools | *Li et al., 2009* | SamTools | This tool is available at: http://samtools.sourceforge.net |
| Software, algorithm | deepTools2 | *Li et al., 2009* | deepTools | This tool is available at: https://deeptools.readthedocs.io/en/develop/ |

*Continued on next page*

*Continued*

| Reagent type (species) or resource | Designation | Source or reference | Identifiers | Additional information |
|---|---|---|---|---|
| Software, algorithm | Integrative Genomics Viewer 2.4.4 | *Robinson et al., 2011* | IGV | This tool is available at: https://software.broadinstitute.org/software/igv/ReleaseNotes/2.4.x |
| Software, algorithm | R version 3.5.1 | R project | R | |
| Software, algorithm | ADS R package | *Pélissier and Goreaud, 2015* | ADS R package | This tool is available at: https://cran.r-project.org/web/packages/ads/index.html |
| Software, algorithm | vbSPT | *Persson et al., 2013* | vbSPT | This tool is available at http://vbspt.sourceforge.net |
| Software, algorithm | Adobe Illustrator CC2017 | Adobe Inc | | |
| Software, algorithm | Prism 7 | GraphPad | | |

## Tissue culture

Human U2OS cells (female, 15 year old, osteosarcoma; STR verified) were cultured at 37°C and 5% $CO_2$ in 1 g/L glucose DMEM supplemented with 10% Fetal Bovine Serum and 10 U/mL Penicillin-Streptomycin, and we subcultivated at a ratio of 1:3 – 1:6 every 2 to 4 days. Stable cell lines expressing the exogenous gene product α-amanitin resistant HaloTag-RPB1(N792D), SNAPf-RPB1(N792D) or Dendra2-RPB1(N792D) were generated using Fugene 6 (Promega) following the manufacturer's protocol, and selection with 2 µg/mL α-amanitin. Stable colonies were pooled and maintained under selection with 1 µg/mL α-amanitin to ensure complete replacement of the endogenous RPB1 pool, as described previously (*Boehning et al., 2018*; *Cisse et al., 2013*). Cells co-expressing SNAPf-RPB1 and Halo-TetR were generated using the previously described SNAP-RPB1 cell line, and transfecting with TetR-HaloTag and a linearized Hygromycin resistance marker using Fugene six following the manufacturer's protocol. Cells were selected and maintained with 100 µg/mL Hygromycin B. Fluorescent cells were selected by labeling the TetR-Halo with 500 nM JF$_{549}$ and using Fluorescence Activated Cell Sorting to identify and keep the fluorescent clones.

Vero cells (*Cercopithecus aethiops* kidney cells; STR verified), were cultured for the growth and propagation of HSV1. Vero cells were cultured at 337°C and 5% $CO_2$ in 4.5 g/L glucose DMEM supplemented with 10% Fetal Bovine Serum and 10 U/mL Penicillin-Streptomycin. Cells were subcultivated at a ratio of 1:3 – 1:8 every 2 to 4 days.

## Virus infection

HSV1 Strain KOS was a generous gift from James Goodrich and Jennifer Kugel (*Abrisch et al., 2015*). UL2/50 was a generous gift from Neal DeLuca (*Dembowski and DeLuca, 2015*). All virus strains were propagated in Vero cells as previously described (*Blaho et al., 2005*). Briefly, cells were infected by incubation at an MOI ~ 0.01 in Medium 199 (Thermo) for 1 hr. 36-48 hpi, cells were harvested by freeze-thawing, pelleted, and sonicated briefly, and then centrifuged to clear large cellular debris. Because we were interested in the early events in infection, approximate titers were first determined by plaque formation assay in Vero cells (*Blaho et al., 2005*). More accurate MOI were determined by infecting U2OS cells plated on coverslips with the same protocol as would be using for imaging experiments. Cells were washed once with PBS, and then 100 µL of complete medium containing 1:10 – 1:10$^5$ dilutions of harvested virus were added dropwise onto the coverslip to form a single meniscus on the coverslip. Infection was allowed to proceed for 15 min at 37 °C. Samples were then washed once with PBS and returned to culturing medium and incubated for 8 hours

before fixation. To measure the MOI, immunofluorescence for the expression of ICP4 using an anti-ICP4 primary antibody (Abcam), and counting the number of infected versus uninfected cells. MOI was then calculated, assuming a Poisson distribution of infection events, as $P\left(k_{inf}\right) = \frac{MOI^{k_{inf}} e^{-MOI}}{k_{inf}!}$, where $k_{inf}$ is the number of infection events per cell. When counting the uninfected cells, this simplifies to $MOI = -\ln\left(f_{uninfected}\right)$. All experiments were performed from the same initial viral stock, with care taken so that each experiment was done with virus experiencing the same total number of freeze/thaw cycles to ensure as much consistency as possible.

### Transient transfection
For experiments where transiently transfected cells were also infected with HSV1, nucleofection was used to achieve more consistent infection across the coverslip. $1 \times 10^6$ cells were trypsinized and resuspended in Kit V buffer plus supplement (Lonza) with 500 ng plasmid, and nucleofected using program X-001, per the manufacturer's instructions. Cells were plated on coverslips and allowed to recover for 48 hr prior to HSV1 infection.

### Live cell imaging
Cells were plated on plasma-cleaned 25 mm circular No. 1.5H cover glasses (Marienfeld High-Precision 0117650) and allowed to adhere overnight. For experiments with HaloTag-expressign cells, cells were incubated with 5–500 nM fluorescent dye (e.g. JF$_{549}$) conjugated with the HaloTag ligand for 15 min in complete medium. Cells were washed once with PBS, and the media replaced with imaging media (Fluorobrite media (Invitrogen) supplemented with 10% FBS and 10 U/mL Penicillin-Streptomycin). For experiments with cells expressing SNAP-RPB1, cells were labeled with 250 nM fluorescent dye (e.g. JF$_{549}$) conjugated with the cpSNAP ligand for 30 min. After labeling, cells were washed for 30 min in complete medium. Prior to imaging, coverslips were mounted in an Attofluor Cell Chamber filled with 1 mL of imaging medium. Cells were maintained at 37°C and 5% CO2 for the duration of the experiment. For long-term time course imaging experiments, cells were plated in 35 mm No. 1.5 glass-bottomed imaging dishes (MatTek), infected with HSV1 at an MOI of ~1, and labeled with JF$_{549}$, and finally the media exchanged for imaging media before placing in a pre-warmed Biostation (Nikon). At 3 hr post infection, infected cells were identified and imaged were taken every 30 s for 5 hr. For phase images, cells were plated and labeled as above, and imaged on a custom-built widefield microscope with a SLIM optics module (PhiOptics) placed in the light path directly before the camera.

### Fluorescence recovery after photobleaching (FRAP)
FRAP experiments were performed as previously described, with modifications. HaloTag-RPB1 cells labeled with 500 nM JF$_{549}$ were imaged on an inverted Zeiss LSM 710 AxioObserver confocal microscope with an environment chamber to allow incubation at 37°C and 5% CO$_2$. JF$_{549}$ was excited with a 561 nm laser, and the microscope was controlled with Zeiss Zen software. Images were acquired with a 63x Oil immersion objective with a 3x optical zoom. 1200 total frames were acquired at a rate of 250 msec per frame (4 Hz). Between frames 15 and 16, an 11-pixel (0.956 μm) circle was bleached, either in the center of a RC, or in a region of the nucleus far from the nuclear periphery or nucleoli.

FRAP movies were analyzed as previously described (*Hansen et al., 2017*). Briefly, the center of the bleach spot was identified manually, and the nuclear periphery segmented using intensity thresholding that decays exponentially to account for photobleaching across the time of acquisition. We measured the intensity in the bleach spot using a circle with a 10 pixel diameter, to make the measurement more robust to cell movement. The normalized FRAP values were calculated by first internally normalizing the signal to the intensity of the whole nucleus to account for photobleaching, then normalizing to the mean value of the spot in the first 15 frames. We corrected for drift by manually updating a drift-correction vector with the stop drift every ~40 frames. FRAP values from individual cells were averaged across replicates to generate a mean recovery curve, and the error displayed is the standard error of the mean.

### Fluorescence loss in photobleaching (FLIP)
FLIP experiments were performed on the same microscope described above for FRAP. Rather than bleach an 11-pixel spot a single time, in FLIP the spot is bleached with a 561 nm laser (or in the case

of Dendra2, photoconverted with a 405 nm laser) between each acquisition frame. Movies were collected for 1000 frames at 250 msec per frame (4 Hz), or one frame per second (1 Hz) for Dendra2.

FLIP movies were analyzed using the same core Matlab code as the FRAP data, except that fluorescence intensities from another 10-pixel circle were recorded to measure the loss of fluorescence elsewhere in the nucleus. This analysis spot was chosen to be well away from the bleach spot, either at a neighboring RC in infected samples or somewhere else in the nucleoplasm far away from both the nuclear periphery and nucleoli. Instead of internally correcting for photobleaching, photobleaching correction was based on an exponential decay function empirically determined to be at a rate of $e^{-0.09}$ per frame. FLIP data from multiple cells were averaged together to determine the mean and standard error for a given condition.

## RNA fluorescence in situ hybridization (FISH) and immunofluorescence (IF)

RNA FISH was used to measure the transcription output for a given RC. To ensure we were measuring nascent transcription, we chose to tile the intronic region of RL2, one of the few HSV1 transcripts with an intron. The 25 oligonucleotide probes were synthesized conjugated with a Cal Fluor 610 dye (Biosearch Technologies; for a full list of oligo sequences see *Supplementary file 1*). FISH was performed based on the manufacturer's protocol. Briefly, cells were plated on 18 mm No. 1.5 coverslips (Marienfield) and infected. At the desired time point, cells were fixed in 4% Paraformaldehyde diluted in PBS for 10 min. After two washes with PBS, coverslips were covered with 70% v/v ethanol and incubated at −20°C for 1 hr up to 1 week.

For hybridizations, coverslips were removed from ethanol and washed in freshly-prepared Wash Buffer A (2 volumes 5x Wash Buffer A, 1 vol formamide, seven volumes $H_2O$) (Bioseach Technologies). Hybridization buffer (10% v/v Dextran Sulfate, 300 mM Sodium Chloride, 30 mM Sodium Citrate, 400, 10% Formamide v/v, and 12.5 nM pooled fluorescent probes) was prepared freshly before each hybridization. A hybridization chamber was prepared with moistened paper towels laid in a 15 cm tissue culture plate. A single sheet of Parafilm was laid over the moistened paper towel. 50 μL of hybridization buffer was pipetted onto the parafilm, and a coverslip inverted into the hybridization buffer. The chamber was sealed with parafilm and placed in a dry 37°C oven for 4–16 hr. After hybridization, coverslips were placed back into a 12-well plate containing 1 mL Wash Buffer A and incubated twice for 20 min in a dry oven at 37°C, with the second wash containing 300 nM DAPI. In a final wash step, cells were washed in Wash Buffer B (Biosearch Technologies). Coverslips were mounted on glass microscope slides in Vectashield mounting medium (Vector Laboratories) and the edges sealed with clear nail polish (Electron Microscopy Sciences). For experiments with combined immunofluorescence and FISH, primary antibody was added to the hybridization buffer at a concentration of 2 μg/mL. An additional wash step with Wash Buffer A containing 1 μg/mL anti-mouse polyclonal antibody conjugated to AlexaFluor 647 was performed before DAPI staining and incubated at 37°C for 20 min.

Samples were imaged on a custom-built epifluorescence Nikon Eclipse microscope equipped with piezoelectric stage control and EMCCD camera (Andor), as well as custom-built filter sets corresponding to the wavelength of dye used. All samples were imaged the same day after hybridaztion and/or incubation with secondary antibody, and all samples to be quantitatively compared across coverslips were imaged on the same day using exactly the same illumination and acquisition settings to minimize coverslip-to-coverslip variation.

## Single particle tracking (spaSPT)

Single particle tracking experiments were carried out as previously described (*Hansen et al., 2017*), but are described here in brief. After overnight growth, U2OS cells expressing Halo-RPB1 were labeled with 50 nM each of $JF_{549}$ and PA-$JF_{646}$. Single molecules imaging was performed on a custom-built Nikon Ti microscope fitted with a 100x/NA 1.49 oil-immersion TIRF objective, motorized mirror are to allow HiLo illumination of the sample, Perfect Focus System, and two aligned EM-CCD cameras. Samples were illuminated using 405 nm (140 mW, OBIS coherent), 561 nm (1 W, genesis coherent), and 633 nm (1 W, genesis coherent) lasers, which were focused onto the back pupil plane of the objective via fiber and multi-notch dichromatic mirror (405 nm/488 nm/561 nm/633 nm quadband; Semrock, NF03-405/488/532/635E-25). Excitation intensity and pulse width were controlled

through an acousto-optic transmission filter (AOTF nC-VIS-TN, AA Opto-Electronic) triggered using the camera's TTL exposure output signal. Fluorescence emissions were filtered with a single band-pass filter in front of the camera (Semrock 676/37 nm bandpass filter). All the components of the microscope, camera, and other hardware were controlled through NIS-Elements software (Nikon).

For all spaSPT experiments, frames were acquired at a rate of 7.5 ms per frame (7 ms integration time plus 0.447 ms dead time). In order to obtain both the population-level distribution of the molecules for masking and the single trajectories, we used the following illumination scheme: First 100 frames with 561 nm light and continuous illumination were collected; then 20,000 frames with 633 nm light at 1–2 ms pulses per frame and 0.4 msec pulses of 405 nm light during the camera dead time; then 100 frames with 561 nm light and continuous illumination were collected. 405 nm illumination was optimized to achieve a mean density of ~0.5 localizations per camera frame, a density sufficiently low to unambiguously identify trajectories, even in dense regions like RCs. Data were collected over multiple courses of infection and 2 to 4 separate days for each condition in order to ensure a sufficiently large sample size.

## ATAC-seq sample preparation

ATAC-seq experiments were performed as previously described (*Buenrostro et al., 2013*). Briefly, 100,000 U2OS cells stably expressing HaloTag-RPB1 were plated and allowed to grow overnight. The following day, cells were infected as described above, and incubated either in complete medium, or complete medium supplemented with 300 µg/mL phosphonoacetic acid (PAA). Infections were timed such that all cells were harvested at once. All the infected cell lines were then trypsinized, and 100,000 cells were transferred to separate eppendorff tubes. Cells were briefly centrifuged at 500 xg for 5 min at 4°C, and the supernatant discarded. After one wash with ice-cold PBS and another 5 min spin at 500 xg and 4°C, cells were resuspended directly in tagmentation buffer (25 µL 2x Buffer TD, 22.5 µL nuclease-free water, 2.5 µL Tn5 (Illumina)) and incubated for 30 min at 37°C. DNA extraction and amplification with barcodes were performed as previously described, with 10–16 total cycles amplification. Barcoded samples were pooled in equimolar amounts and sequenced using a full flow-cell of an Illumina Hi-Seq 2500 per replicate. Three replicates were performed, although the first replicate was deemed to have been over-amplified during the PCR step, and thus was omitted from the analysis.

## Oligopaint on infected cells

For DNA FISH experiments, custom pools of fluorescently labeled DNA oligos were generated using previously published protocols (*Boettiger et al., 2016*). Briefly, oligo sequences tiling a 10,016 bp region in the Unique Long arm (JQ673480 position 56,985 to 66,999) and a 7703 bp region in the Unique Short arm (JQ673480 position 133,305 to 141,007) were manually curated using oligo BLAST (NCBI) against the HSV1 and human genomes with the following settings, following guidelines for Tm, GC-content, and length from previous Oligopaint protocols (*Boettiger et al., 2016*). Individual oligos were purchased commercially (the sequences for these oligos can be found in *Supplementary file 2* and pooled. PCR was used to introduce a common T7 promoter on the 3' end of the final probe sequence, then the PCR products were gel purified before in vitro transcription to generate ssRNA complimentary to the hybridization sequence. Finally, the entire RNA pool was reverse transcribed in a single reaction using Maxima RT (ThermoFisher) using either AlexaFluor-647 or AlexaFluor-555 5'-labeled oligos as the reverse transcription primer. After acid hydrolysis to remove the RNA, oligos were purified using high binding capacity oligo cleanup columns (Zymo) and resuspended in TE.

Cells were plated on 18 mm coverslips and infected as described above. Infected was allowed to progress for between 3 and 8 hr in the presence or absence of phosphonoacetic acid, then fixed with 4% paraformaldehyde for 15 min. Coverslips were washed twice with PBS, then incubated with 100 mM Glycine in PBS for 10 min. Samples were permeabilized for 15 min with 0.5% Triton-X100 in PBS, then washed twice with PBS. After permeabilization, samples were treated with 100 mM HCl for 5 min, then washed twice with PBS. Prior to hybridization, samples were washed twice with 2X SSC (300 mM NaCl, 30 mM Sodium Citrate), and then incubated at 42°C for 45 min in 2X SSC with 50% v/v Formamide. Coverslips were inverted onto a slide containing 25 µL hybridization buffer (300 mM NaCl, 30 mM Sodium Citrate, 20% w/v Dextran Sulfate, 50% v/v Formamide, and 75 pmol of

fluorescently labeled oligos) and sealed with rubber cement. Samples were denatured at 78°C on an inverted heat block for 3 min, then incubated in a humidified chamber at 42°C for 16 hr. Samples were then removed from the glass slides and washed twice to 60°C with pre-warmed 2x SSC for 15 min, then washed twice with 0.4x SSC at room temperature for 15 min. Finally, coverslips were mounted on glass slides with Vectashield mounting medium.

DNA FISH samples were imaged on the same microscope as described above for immunofluorescence and RNA FISH. Z-stack images were collected from all the way below the focal plane to all the way above the focal plane, with a step size of 100 nm. All samples were imaged on the same day using the same illumination and acquisition settings to minimize coverslip to coverslip differences.

## PALM of Pol II in RCs

For PALM experiments to precisely localize Pol II molecules within RCs, cells were labeled with 500 nM PA-JF$_{549}$, and then infected as described above. Cells were fixed in 4% Paraformaldehyde in PBS, washed twice with PBS. Fluorescent 100 nm and 200 nmTetraspek beads were mixed in a 9:1 ratio then diluted 1000-fold in PBS. 100 µL was added to each coverslip and allowed to settle for 5 min, followed by 5 min of washing while rocking. Coverslips were mounted in Attofluor Cell Chambers and covered with PALM imaging buffer (50 mM NaCl, 50 mM Tris pH 7.9, 2 mM Trolox) to reduce triplet-state blinking.

Samples were imaged on a custom-built Nikon Ti microscope equipped similarly to the microscope for single particle tracking, with some differences described here. An Adaptive Optics module (MicAO) and a removable cylindrical lens were placed in the light path ahead of the EM-CCD (Andor iXon Ultra 897) cameras in the left and right camera ports (respectively) of the microscope. Astigmatism for precise 3D localization was introduced using the Adaptive Optics system. The Adaptive Optics system was controlled through the MicAO software and calibrated on 200 nM Tetraspek beads based on the total photon yield and point spread function shape after iterative tuning of the deformable mirror. After optimization, a slight astigmatism in the vertical Zernike mode (Astigmatism 90°=0.060) was added, and several z-stacks of 100 nM Tetraspek beads with 10 nm between slices to calibrate the PSF shape with the Z-position. 30,000 frames were acquired with the 561 nm laser line and increasing amounts of 405 nm illumination in order to keep the number of single molecules consistent across the duration of acquisition.

## STORM on infected cells

For STORM experiments to visualize both RNA Polymerase II and the viral DNA, U2OS cells stably expressing Halo-RPB1 were plated on coverslips, labeled with 300 nM JF$_{549}$, and infected with the UL2/50 virus strain (*Dembowski and DeLuca, 2015*) as described above. After infection incubation with virus, cells were transferred into complete medium containing 300 µg/mL PAA for two hours to prevent replication. After two hours, cells were released from inhibition by exchanging the culture medium with complete medium containing 2.5 µM 5-Ethynyldeoxyuridine for 4 hr. Cells were fixed with 4% Paraformaldehyde in PBS for 10 min, then permeabilized with 0.5% Triton X100 in PBS for 10 min. Copper(1)-catalyzed alkyne-azide cycloaddition was performed with the ClickIT imaging kit following the manufacturer's protocol (Thermo). Coverslips were mounted in Attofluor Cell Chambers and covered with freshly-made STORM buffer (50 mM NaCl, 50 mM Tris pH 7.9, 10% D-glucose, 10 mM DTT, 700 µg/mL Glucose Oxidase (Sigma), and 4 µg/mL catalase). STORM experiments were performed on the same microscope described for PALM.

## IUPred disorder prediction

Disorder predictions were preformed using a custom built python script to implement the IUPred intrinsic disorder prediction program (*Dosztányi et al., 2005a*; *Dosztányi et al., 2005b*). Specific protein sequences were placed in a table and this was fed into the script. All protein sequences were downloaded from the reference organism at uniport.org. The resulting traces were smoothed by a rolling mean of 8 residues to remove noise and prevent single low-energy residues from splitting single large IDRs into multiple apparent IDRs. Contiguous substrings of residues with centered-mean IUPred disorder likelihood greater than 0.55 were annotated as 'disordered regions' (*Figure 1E*), and those contiguous regions larger than 10 amino acids were included in the calculation of 'fraction IDR'.

## spaSPT data processing

SPT data sets were processed in four general steps using a custom-written Matlab (Mathworks): 1) Masks for RCs were annotated manually, 2) the masks were corrected for drift throughout the sample acquisition, 3) particles were localized and trajectories constructed, and 4) trajectories were sorted as 'inside' compartments or 'outside'.

First, the 100 frames at the beginning and the end of each movie were separately extracted and a maximum-intensity projection used to generate 'before' and 'after' images of the cell or cells in the field of view. These images would be used to correct for movement of the cell as well as the individual RCs. For each cell, the nucleus was annotated in the 'before' image, and then again in the 'after' image. We assumed that the cell movement over the ~4 min of acquisition was approximately linear and calculated the drift-corrected nuclear boundary for every frame in the stack of SPT images. The same procedure was applied to each of the replication compartments. Particle localization and tracking were implemented based on an adapted version of the Multiple Target Tracking (MTT) algorithm, available at https://gitlab.com/tjian-darzacq-lab/SPT_LocAndTrack(**Hansen, 2019**; copy archived at https://github.com/elifesciences-publications/SPT_LocAndTrack). In the first step, particles were identified with the following input parameters: Window = 9 px; Error Rate = $10^{-6.25}$; Deflation Loops = 0. Following detection, a mask generated from the drift-corrected nuclear boundary was applied to discard any detections not within the nucleus. Trajectories were reconstructed with the following parameters: Dmax = 10 $\mu m^2$/sec; Search exponent factor = 1.2; Max number of competitors = 3; Number of gaps allowed = 1.

Finally, after trajectories have been reconstructed, they were sorted as 'inside' RCs or 'outside'. To minimize the potential for bias in calling trajectories inside of compartments, we only required a single localization in a trajectory to fall within a compartment for that trajectory to be labeled as 'inside'. As is discussed in the main text, we tested this sorting strategy for implicit bias by computationally generating mock RCs in uninfected or infected samples (**Figure 2—figure supplement 3**). To do this, all the annotations for RCs from the infected samples (n = 817), as well as the distribution of number of RCs per infected cell, were saved in a separate library. We then took the uninfected cells and, in a similar process as described above, annotated the nuclear boundary and nucleoli. We then randomly sampled from distribution of RCs per cell a number of RCs to place in the nucleus, and then from the library of annotations randomly chose these RCs and placed them in the nucleus by trial-and-error until all of the chosen RCs could be placed in the nucleus without overlapping with each other, a nucleolus, or the nuclear boundary (**Figure 2—figure supplement 3A**). The SPT data were then analyzed as above—drift-correction, followed by localization, building of trajectories, and sorting into compartments—using the exact same parameters. We also followed this same procedure of randomly choosing and placing artificial RCs in infected cells, this time avoiding previously annotated RCs instead of nucleoli **Figure 2—figure supplement 3B**.

## Two-state kinetic modeling using Spot-On

We employed the Matlab version of Spot-On (available at https://spoton.berkeley.edu) in our analysis and embedded this code into a custom-written Matlab routine. All data for a given condition were merged, and histograms of displacements were generated for between 1 and 7 Δt. These histograms were fitted to a two-state kinetic model which assumes one immobile population and one freely diffusing population: Localization Error = 45 nm; $D_{free}$ = [0.5 $\mu m^2$/s, 25 $\mu m^2$/s]; $D_{bound}$ = [0.0001 $\mu m^2$/s, 0.08 $\mu m^2$/s]; Fraction Bound = [0, 1]; UseWeights = 1; UseAllTraj = 0; JumpsToConsider = 4; TimePoints = 7; dZ = 0.700. Trajectory CDF data were fit to a two-state model as first outlined by Mazza and colleagues, then expanded with implementation in Hansen and colleagues.

Spot-On has been shown to robustly estimate allthe fitted parameters, provided there is sufficient data—at a minimum 1000 trajectories for a 2-state fit of a model protein with diffusion characteristics similar to Pol II (50% bound, Dfree = 3.5 $\mu m^2$/sec) (**Figure 2—figure supplement 1A**) (**Hansen et al., 2018**). Because of the sparsity of the data we collected per cell, we found that we could not reliably generate single-cell statistics, particularly within RCs where the total number of trajectories per cell fell well below the 1000-trajectory threshold (**Figure 2—figure supplement 1B**). In order to robustly fit our data and simultaneously estimate its variability, we first calculated the number of cells we would need to confidently fit all compartments and found 15 cells to optimal (**Figure 2—figure supplement 1B**). We then implemented a random subsampling approach where

15 cells from a particular condition were randomly chosen and analyzed. The $D_{free}$, $D_{bound}$, and Fraction Bound were calculated iteratively for trajectories inside and outside of RCs. This random resampling was repeated 100 times, and the median values and standard deviations calculated and reported. When compared to the values that would have been obtained for taking the mean and standard deviation of the individual biological replicates, our subsampling approach agreed with these means within the measurement error (*Figure 2—figure supplement 1C*).

## Analysis of angular distribution

Angular distribution calculations were performed using a custom written routine in Matlab, implementing a previous version of this analysis (available at https://gitlab.com/anders.sejr.hansen/anisotropy; *Hansen, 2018*, copy archived at https://github.com/elifesciences-publications/anisotropy/). To analyze the angular distribution of trajectories in different conditions, we started with the list of trajectories generated above, annotated as either 'inside' or 'outside' of RCs. A trajectory of length N will have N-2 three-localization sets that form an angle, and so we built a matrix consisting of all consecutive three-localization sets. It is crucially important that only diffusing molecules be considered in the analysis, as localization error of bound molecules would skew all of the data to be highly anisotropic. To address this, we used two criteria. First, we only applied a Hidden-Markov Model based trajectory classification approach to classify trajectories as either diffusing or bound (*Persson et al., 2013*), and kept only the trajectories that were annotated as diffusing. Second, we applied a hard threshold that both translocations (1 to 2, 2 to 3) had to be a minimum of 150 nm, which ensured that we could accurately compute the angle between them. Because a particle may diffuse into or outside of the annotated region, we counted a trajectory as 'inside' only if the vertex of the angle occurred within an annotated region.

## ATAC-seq analysis

Sequenced reads were mapped separately to hg19 genome using Bowtie2 (*Langmead and Salzberg, 2012*) with the following parameters: `–no-unal –local –very-sensitive-local –no-discordant –no-mixed –contain –overlap –dovetail –phred33`. Reads were separately mapped to the HSV1 genome, JQ673480, using Bowtie2 with the following parameters: `–no-unal –no-discordant –no-mixed –contain –overlap –dovetail –phred33`. The bam files were converted to bigwig files and visualized using IGV (*Robinson et al., 2011*). TSS plots were generated using Deeptools suite (bamCoverage, computeMatrix, plotHeatmap tools) using UCSC TSS annotations for hg19 genome (*Ramírez et al., 2016*), and using a highly refined map of the gene starts in HSV1 kindly provided by Lars Dölken (University of Cambridge, to be published separately).

## Analysis of immunofluorescence, RNA, and DNA FISH

All cells were analyzed using a custom Matlab script. First, a single image for each color channel was generated by automatically identifying the focal plane of the stack, and then integrating the pixel intensity for all pixels 1 μm above and below the focal plane. Nuclei were automatically segmented, but replication compartments could not reliable by detected using simple thresholding, and so each was manually annotated. A region of the image was selected to represent the black background, and the mean pixel value of this region was subtracted from every pixel in the image. After segmentation, the pixel values for each nucleus were recorded, as well as every RC within a given nucleus, and these were used to measure the signal within the RC, as well as the fraction of signal within compared to the rest of the nucleus (immunofluorescence only).

## Quantification of DNA content within RCs

DNA FISH data were compared with ATAC-seq data for the six hpi timepoint. Despite the fact that U2OS are hypertriploid, we based all the calculations on the DNA content of a diploid cell. As such, the values presented here likely represent an upper bound on the relative concentrations of host and HSV1 gDNA for our experiments. Volume estimates for nuclei were based on data from *Monier et al. (2000)*; volumetric measurements for RCs were taken directly from the annotations of the DNA FISH data.

## PALM spatial statistics

Spatial statistics were collected on cells using previously published methods (*Boehning et al., 2018*). First, cell boundaries and replication compartments were annotated as for spaSPT experiments (above). Particularly for small objects like RCs, edge correction is crucial for accurate spatial point pattern statistics. Given a set of detections P, we used the estimator $f$ to correct for biases generated by points near the RC boundary:

$$f(i,j,r) = \begin{cases} 0, & if\ d(i,j) > r \\ \frac{2\pi\ d(i,j)}{C_{in}}, & otherwise \end{cases}$$

where $d(i,j)$ is the distance between points $i$ and $j$ for $i,j \in P$, and $C_{in}$ is arclength of the part of the circle of $d(i,j)$ centered on $i$ which is inside the annotated region (*Goreaud and Pélissier, 1999*). We then calculated N(r), the local neighborhood density:

$$N(r) = \frac{1}{N_p} \sum_{i \in P} \sum_{i \neq j} f(i,j,r)$$

where $N_p$ is the total number of detections within the region (*Goreaud and Pélissier, 1999*).

The modified L-function is compared to complete spatial randomness (CSR), a homogenous Poisson process with intensity $\lambda$, equal to the density of detections in the region of interest A. The K-Ripley function is defined as:

$$K(r) = \frac{N(r)}{\lambda}$$

(*Ripley, 1977*). We estimated the modified L-function given by:

$$L(r) - r = \sqrt{\frac{K(r)}{\pi}} - r$$

(*Goreaud and Pélissier, 1999*). For the modified L-function, a spatial distribution with CSR remains at 0 for all radii. To implement this analysis, we used a previously published python script and the ADS R package to estimate the spatial statistics (*Boehning et al., 2018*; *Pélissier and Goreaud, 2015*). In order to estimate the error in our measurements, for each cell we performed random subsampling of the data, before annotation, to randomly select 25,000 detections 100 times, and fed these subsampled data to the R script computing the statistic. For very small radii, a high L(r)-r value is likely due to blinking and other photo-physical artifacts (*Annibale et al., 2011*), but at length scales larger than localization error the method becomes robust.

## Data and software availability

The GEO accession number for the ATAC-seq data is: GSE117335. The SPT trajectory data are available via Zenodo at DOI: 10.5281/zenodo.1313872. The software used to generate these data is available at https://gitlab.com/tjian-darzacq-lab.

## Acknowledgements

We thank James Goodrich, Jennifer Kugel, and Robert Abrisch for providing the HSV1 strain KOS that began this project, and for helpful discussions. Thank you also to Stephen Rice for the n504 and n406 HSV1 strains, and Neal DeLuca for the UL2/50 HSV1 strain. Thank you to Luke Lavis for generously providing all of the Janelia Fluor dyes that enabled these experiments. Thank you to Ana Robles, Mustafa Mir, and Astou Tangara for their tireless work keeping the microscopes in working order. Thank you to all of the individuals who provided reagents, comments, and critical insight for this manuscript, including Claudia Cattoglio, Shasha Chong, Thomas Graham, Britt Glaunsinger, Ella Hartenian, Matthew Parker, James McSwiggen, and the Tjian and Darzacq Lab members. This work was supported by NIH grants UO1-EB021236 and U54-DK107980 (XD), the California Institute of Regenerative Medicine grant LA1-08013 (XD), by the Howard Hughes Medical Institute (003061, RT). A.B.H. is supported by the NIH predoctoral fellowship T32 GM098218. Portions of this work were performed on shared instrumentation at the CRL Molecular Imaging Center, supported by The

Gordon and Betty Moore Foundation. We would like to thank Holly Aaron and Jen-Yi Lee for their assistance. DNA sequencing in this work used the Vincent J Coates Genomics Sequencing Laboratory at UC Berkeley, supported by NIH 669 S10 Instrumentation Grants S10RR029668 and S10RR027303.

## Additional information

### Competing interests

Robert Tjian: is one of the three founding funders of *eLife*, and a member of *eLife's* Board of Directors. The other authors declare that no competing interests exist.

### Funding

| Funder | Grant reference number | Author |
|---|---|---|
| National Institutes of Health | UO1- 497 EB021236 | David Trombley McSwiggen<br>Anders S Hansen<br>Yvonne Hao<br>Alec Basil Heckert<br>Kayla K Umemoto<br>Claire Dugast-Darzacq<br>Xavier Darzacq |
| National Institutes of Health | U54-DK107980 | David Trombley McSwiggen<br>Anders S Hansen<br>Yvonne Hao<br>Alec Basil Heckert<br>Kayla K Umemoto<br>Claire Dugast-Darzacq<br>Xavier Darzacq |
| Howard Hughes Medical Institute | 003061 | David Trombley McSwiggen<br>Anders S Hansen<br>Sheila S Teves |
| California Institute for Regenerative Medicine | LA1-08013 | Anders S Hansen<br>Alec Basil Heckert<br>Xavier Darzacq |
| National Institutes of Health | K99GM130896 | Anders S Hansen |

The funders had no role in study design, data collection and interpretation, or the decision to submit the work for publication.

### Author contributions

David Trombley McSwiggen, Conceptualization, Resources, Software, Formal analysis, Supervision, Investigation, Visualization, Methodology, Writing—original draft, Writing—review and editing; Anders S Hansen, Sheila S Teves, Conceptualization, Software, Writing—review and editing; Hervé Marie-Nelly, Resources, Software, Writing—review and editing; Yvonne Hao, Kayla K Umemoto, Investigation, Writing—review and editing; Alec Basil Heckert, Software, Writing—review and editing; Claire Dugast-Darzacq, Resources, Writing—review and editing; Robert Tjian, Xavier Darzacq, Conceptualization, Supervision, Funding acquisition, Writing—review and editing

### Author ORCIDs

David Trombley McSwiggen (iD) http://orcid.org/0000-0003-3844-7433
Anders S Hansen (iD) http://orcid.org/0000-0001-7540-7858
Sheila S Teves (iD) http://orcid.org/0000-0002-1220-2414
Alec Basil Heckert (iD) http://orcid.org/0000-0001-8748-6645
Claire Dugast-Darzacq (iD) http://orcid.org/0000-0001-8602-3534
Robert Tjian (iD) https://orcid.org/0000-0003-0539-8217
Xavier Darzacq (iD) http://orcid.org/0000-0003-2537-8395

Decision letter and Author response
Decision letter https://doi.org/10.7554/eLife.47098.033
Author response https://doi.org/10.7554/eLife.47098.034

## Additional files

### Supplementary files

• Supplementary file 1. Fluorescent oligonucleotide sequences for RNA fluorescence in situ hybridization.
DOI: https://doi.org/10.7554/eLife.47098.023

• Supplementary file 2. DNA oligonucleotide sequences for oligopaint.
DOI: https://doi.org/10.7554/eLife.47098.024

• Transparent reporting form
DOI: https://doi.org/10.7554/eLife.47098.025

### Data availability

The GEO accession number for the ATAC-seq data is: GSE117335. The SPT trajectory data are available via Zenodo at DOI: 10.5281/zenodo.1313872. The software used to generate these data is available at https://gitlab.com/tjian-darzacq-lab/SPT_LocAndTrack (copy archived at https://github.com/elifesciences-publications/SPT_LocAndTrack) and https://gitlab.com/anders.sejr.hansen/anisotropy (copy archived at https://github.com/elifesciences-publications/anisotropy).

The following datasets were generated:

| Author(s) | Year | Dataset title | Dataset URL | Database and Identifier |
|---|---|---|---|---|
| McSwiggen DT, Hansen AS, Teves S, Marie-Nelly H, Hao Y, Heckert AB, Umemoto KK, Dugast-Darzacq C, Tjian R, Darzacq X | 2018 | Relative accessability of HSV1 genomic DNA compared with its host cell (ATAC-seq) | https://www.ncbi.nlm.nih.gov/geo/query/acc.cgi?acc=GSE117335 | NCBI Gene Expression Omnibus, GSE117335 |
| McSwiggen DT, Hansen AS, Teves S, Marie-Nelly H, Hao Y, Heckert AB, Umemoto KK, Dugast-Darzacq C, Tjian R, Darzacq X | 2018 | Single Particle Tracking data for U2OS cells after infection | http://doi.org/10.5281/zenodo.1313872 | Zenodo, 10.5281/zenodo.1313872 |

The following previously published dataset was used:

| Author(s) | Year | Dataset title | Dataset URL | Database and Identifier |
|---|---|---|---|---|
| Hansen AS, Woringer M, Grimm JB, Lavis LD, Tjian R | 2017 | Simulated data for 'Spot-On: robust model-based analysis of single-particle tracking experiments' | http://doi.org/10.5281/zenodo.835541 | Zenodo, 10.5281/zenodo.835541 |

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
