## [Decision Letter]

Thank you for submitting your article "DNA-mediated Nuclear Compartmentalization Distinct From Phase Separation" for consideration by *eLife*. Your article has been evaluated by Kevin Struhl (Senior Editor) and Jessica Tyler (Reviewing Editor). Dr Tyler's review is appended below, and the content of this decision letter also reflects input that she obtained from a leading expert in the liquid:liquid phase separation field and a leading expert on HSV-1 genome function.

Summary:

Both editors and external experts agreed that the work is timely, important and of high quality, thoughtfully addressing a variety of different hypotheses for why RNA polymerase II is enriched in membraneless compartments that contain the lytic Herpes Simplex Virus -1 (HSV-1) genome. We all agree that you fairly interpret your results and do a good job of explaining how your data goes against the simple predictions of liquid-liquid phase separated states. Furthermore, this work is congruent with, and adds important insights into, our understanding of the biology of HSV-1 and function of DNA binding proteins.

All four of the consulted experts/editors had access to the reviews from a previous journal that you provided with your submission, and your responses to the concerns raised in these previous reviews. The *eLife* editors and consulted experts all felt that your responses were appropriate and adequately addressed the previous reviewer comments. The problem (as you also argue in your responses to the previous journal's reviews) is that the field does not have good definitions for how to define the criteria for what an LLPS compartment should behave like in vivo. We think this is exactly the type of work that is needed to catalyze a deeper discussion in the field.

Taking this into account, the reviewers agreed that the work should be published largely as is, with the following suggestion: please consider softening the title to either "Evidence for DNA-mediated Nuclear Compartmentalization Distinct From Phase Separation", or "Is there DNA-mediated Nuclear Compartmentalization Distinct From Phase Separation?"

Our detailed comments are appended below for reference:

The compartmentalization of proteins into non-membrane bound, nuclear substructures is an important phenomenon that contributes to a multitude of molecular processes. An understanding of the mechanisms underlying various nuclear compartments is a burgeoning field. In particular, there is a growing and strong appreciation in liquid-liquid phase separation (LLPS) as an important feature of some nuclear compartments. In this report, McSwiggen et al. test whether LLPS is involved in the compartmentalization of replication centers (RCs), which are large nuclear compartments established upon the infection of Herpes Simplex Virus. RCs are composed of viral DNA that sequester host proteins important for the transcription of the viral genome, notably RNA polymerase II (RNAP II). The authors find that RC display characteristics distinct from traditional LLPS. To this end, the authors use live imaging analyses to track RNAP II movement within both the RCs and the nucleoplasm. They find that RNAP II enters and leaves the RC with similar kinetics, suggesting that RNAP II is not beholden to restricted diffusion characteristic of LLPS. Although the findings of this paper do not entirely rule out the possibility of a role for LLPS in nucleating the RC, the authors provide evidence of an alternative mechanism that concentrates RNAP II to the RC based on non-specific binding of RNAP II to the nucleosome-free viral genome. They show that the viral genome is predisposed for non-specific binding due to its high accessibility. Moreover, they demonstrate that other non-specific DNA binders, such as TetR and LacI, can similarly concentrate within the RCs. This latter experiment is particularly powerful in showing that non-nucleosomal DNA is sufficient for sequestering DNA-binding proteins and creating a unique biochemical environment in the cell. In sum, the findings of this paper are important to furthering the understanding of the dynamics of non-membrane bound nuclear substructures. Furthermore, this work has important implications for DNA binding factor localization, chromatin function and HSV-1 biology.

---

## [Author Response]

[Editors' note: we include below the reviews obtained from another journal, along with the authors’ responses.]

The reviewers’ comments from our initial submission were largely quite positive. Reviewer 1 had many criticisms, though many of these stemmed from a misunderstanding of the experiments or of the model we are proposing. Reviewer 2 was much more optimistic and suggested some additional experiments for clarity. Reviewer 3 was also quite positive, though they telegraph a significant bias favoring phase separation as a mechanism in their review.

Reviewer #1:

In this manuscript by McSwiggen et al., the authors use an array of cutting-edge imaging and ATAC-seq approaches to study Pol II exploration of herpesvirus genomes that reside in replication compartments (RCs). While the approaches are impressive and the concept interesting, there are many sweeping conclusions drawn largely from drug experiments that lack proper controls. In addition, there is little regard for the complexity of RCs or the underlying biology of infection that offer alternatives, which seem to be brushed aside without rigorous testing. In addition, there is little to no mechanistic insight into the events being imaged. The choice of cell line is also a concern.Major Concerns:As the authors state, they use a "battery" of live cell imaging approaches. However, each approach seems to draw a sweeping conclusion that dismisses other possibilities in order to move to the next, rather than solidifying the basis for each conclusion, becoming more of a "bewildering battery".

We respectfully disagree with the statement that our conclusions are sweeping, or that they are made at the expense of other parsimonious explanations. For the few instances where Reviewer 1 has offered an alternative explanation, they are addressed below. We also are open to other specific suggestions that might challenge our current model, if such suggestions arise. We appreciate that explanations for our conclusions may have been couched too much in the parlance of particular imaging techniques, and we have tried to improve the clarity for a more general audience.

In addition, in each case a drug is used but not properly controlled, and in many experiments critical controls for probe specificity are missing. A good example is the PAA data presented in Figure 4. The authors assume PAA treatment worked, but they need to perform some kind of late gene expression analysis to confirm this [sic.].

We believe that we have performed all of the pertinent and reasonable controls with respect to each of our chemical perturbation experiments, and have included these data in the manuscript in as transparent a manner as possible. In the case of phosphonoacetic acid (PAA) treatment, the concentration of 300 µg/mL is widely used to prevent HSV1 replication, even in U2OS cells (Boissière et al., 2004; Crumpacker, 1992; Hancock et al., 2010; Lee et al., 2016; Rutkowski et al., 2015; Scott et al., 2001; Wyler et al., 2017). Also, we have direct experimental evidence showing that our protocol for treating infected cells with HSV1 indeed prevents viral replication, not the least of which is the fact that cells infected for 6 hours in the presence of PAA show two orders of magnitude lower fluorescence signal for DNA than cells infected for the same amount of time without addition of PAA (Figure 4C). As far as an assay goes, this is much stronger evidence than indirect assays such as qPCR of Late genes.

Indeed, a major problem with the paper is that it is almost entirely imaging with no real validation.

We reject the notion that data and conclusions generated via microscopy are any less valid than bulk biochemistry techniques, so long as one has the proper controls—controls that we have carefully performed and included. One could argue that the live cell and fixed cell imaging experiments are the best controls for the multitude of biochemical experiments so widely employed in the virus field and certainly should serve as a good complement to the more conventional strategies which have been informative but also necessarily limited in some cases. As is often the case, when new advanced technologies are applied to old problems, they may reveal entirely new mechanisms that have been invisible and this is especially true with single molecule, single cell quantitative imaging strategies. We hope it is evident that science moves forward best when orthologous methods are applied and new information interpreted without prejudice.

In addition, early in paper the authors state that low MOI is used, although in the methods an MOI of 1 is mentioned, yet both cells in the field of view in Figure 4B are infected and both contain multiple puncta, so they must be infected at quite high MOI if PAA blocked DNA replication as assumed

The term “low” is used in our manuscript to contrast our experiments to those by previous groups which were performed at MOI of 10 or higher (Chang et al., 2011; Taylor et al., 2003), but we concede that this is a relative term that is irrelevant to the remainder of the manuscript. As such, we’ve removed this specific language.

Two points are essential to consider here: First, MOI is a statistic of the mean number of infectious particles (as measured, most commonly, by plaque assay or similar) the average cell in a culture will receive upon addition of a virion-containing solution. The number of infectious particles a given cell receives is best modeled by a Poisson Random process (P(k) = e^-m^m^k^/k!, where k is the number of infectious viral particles per cell, and m is the MOI), and for an MOI of 1 it should follow the theoretical distribution shown in Author Response Image 1. Thus, approximately 1/3 of the cells will be uninfected, 1/3 will receive 1 infectious particle, 1/6 will receive two, and so on. This makes it highly probable that more than one infected cell will appear in a single field of view.

Second, almost all measures of MOI, especially ones such as a plaque assay, measure only the number of fully-infectious particles, as they assay new virion production as the end-point. As the reviewer points out later in their comments, not every viral particle that enters a cell will cause productive lytic infection. In our case, where we are infecting with an MOI of 1—calculated using the fraction of uninfected cells as measured by ICP4+ immunofluorescence at 4 hours post infection—we find that the infected cells have multiple RCs. Regardless, these defective virions still have genomes that replicate and that are transcribed by Pol II, as we have measured and demonstrated using time lapse imaging and RNA FISH respectively, and so we feel it is perfectly reasonable to conduct our analyses under these conditions.

However, there is no uninfected control to show these FISH puncta are even specific. As such, all of the analysis stemming from this is flawed.

Uninfected cells were not shown in the main figure to conserve figure space. We have now included full fields of view in Figure 6—figure supplement 1 where uninfected cells can be clearly seen.

There are similar problems with many other figures. For example, the major conclusion that IDRs in either host or viral proteins are not involved is based on the assumption that viral RCs are accessible to the drugs used and that the drugs work against viral proteins. Is there evidence for this?

In Figure 3 we directly measured the nascent transcription of the ICP0 transcript as a means of confirming that both Flavopiridol and Triptolide inhibit transcription. Given that these drugs are quite low in molecular weight (~480 and 360 Da, respectively), and that they are permeable to the cell membrane, it would be difficult to imagine a scenario where they were unable to access Pol II inside of the RCs. We make no claims that these drugs work against any viral proteins; rather we show that they are very effective at preventing new Pol II-mediated transcription within RCs.

Confounding points about drug activity (e.g. outside RCs in Figure 3D) are ignored as "some resistance" when it is essentially complete resistance.

Host gene transcription is severely reduced and modified due to the viral infection and we therefore do not discuss drug activity on host genes. Once again, we have proper controls to show that the transcriptional inhibition is robust and efficient on viral genes, which is essential and sufficient for our conclusion. The fact that the fraction of bound Pol II outside of RCs hovers around ~30%, even after transcription inhibition, is an unexpected result that we have insufficient data to fully explain. However, we have some ideas based on our observations as well as on previously published work. It is known that although the majority of transcription in the host chromatin is redirected to the virus, some host transcription remains. For these transcripts, the polymerase experiences a defect in proper termination, and so can run on hundreds of kilobases past its normal termination site (Rutkowski et al., 2015). In addition, we can see from the ATAC-seq data in our own experiments that regions around the TSS become more accessible. We see a broader peak of accessibility in intra-nucleosome sized fragments, and smaller peaks corresponding to well-positioned nucleosomes in the mononucleosome sized fragments in Figure 4G at 4 and 6 hours post infection. This increase in accessibility could lead to more transient Pol II–DNA binding in much a similar manner as it does in the RC.

We have no data to say whether this 30% bound fraction is bound stably to host chromatin or whether it is transiently bound as we observe inside of RCs. While interesting, this phenomenon is outside the scope of our claims for this paper, and lacking any significant evidence that it is caused by a given mechanism, we acknowledge it in the manuscript but do not wish to speculate any further so as to not over interpret our data.

It is notable that the drugs in question do not affect ICP8 localization to RCs, and ICP8 has been shown to bind PolII. Points like this are ignored but could easily explain many of the observations; are the IDR drugs ineffective in disrupting ICP8-PolII interaction? Some form of pulldown and greater exploration of this is needed, beyond just imaging with assumptions.

We thank the reviewer for this comment. We had no intention to suggest that Pol II only interacts with the viral DNA: It is well known that a number of viral proteins interact with Pol II or other members of the core Pre-Initiation Complex. This includes the viral proteins VP16, ICP4, ICP8, ICP27, and ICP22 among others. It is certainly the case that each of the above proteins serve important, often indispensable functions in the biology of the virus. Both ICP8 and ICP27 have been shown to Co-IP with each other and with RNA Polymerase II in an ICP27-dependent manner (Zhou and Knipe, 2002). If fact, other reports have suggested what reviewer 1 is suggesting: That ICP27 is directly responsible for recruiting Pol II to RCs (Dai-Ju et al., 2006). This paper reports viral mutants of ICP27 (called n504 and n406) which they claim form RCs but do not recruit Pol II. Naturally, we were interested in these viral strains, as this would offer the simplest explanation for Pol II recruitment.

Unfortunately, we found that we were completely unable to replicate these authors’ data regarding Pol II recruitment. These mutants abrogate the interaction between ICP8 and Pol II (Zhou and Knipe, 2002), but what we found is that despite a deficiency in activating replication, once RCs are formed, Pol II was recruited to RCs as robustly as with the WT virus (Figure 3—figure supplement 1). Furthermore, FRAP of cells infected with either n406 or n504 have recoveries indistinguishable from WT virus. We had not included this data in the manuscript because it might distract from our key findings, and because we did not wish to create animosity with other groups where avoidable, but we now see that this is an important distinction with relevance given previous literature and have since included it.

Other evidence that alternative proteins are not generally responsible for the recruitment of Pol II can be seen in our other data. Specifically, Pol II does not show a different diffusion coefficient in RCs, as one might predict if there were changes in the size of the complex. Further, the FLIP data presented in Figure 2F shows that the rate of unbinding of Pol II is actually slightly faster inside of RCs, which argues against the formation of a stable complex as one might expect if Pol II were stably interacting with and predominantly recruited via ICP8, ICP27, or an as-yet-unknown viral factor. As the reviewer points out, we cannot say for sure that 1,6-HD disrupts ICP27-Pol II interactions, but taken together with the above arguments, we believe that we have sufficient evidence to disfavor ICP8/ICP27 interactions as a major contributor to bulk recruitment of Pol II to RCs.

Why are so many tegument proteins included in the IDR analysis in Figure 1? They are structural proteins and may skew this analysis in favor of the theory. The analysis should be limited to proteins that function in transcription.

It is incorrect to say that all tegument proteins are structural. Tegument proteins are part of the payload delivered to the cell upon viral entry past the cell membrane. Important viral transcriptional activators—particularly the viral protein VP16 that is known to be a strong driver of Pol II-mediated transcription for the Immediate Early and Early genes—are among the list of tegument proteins. We group these together with the Immediate Early genes, as we say in the paper, to represent a category of proteins that include the collective set of polypeptides available to the virus to initiate the formation of RCs.

This leads to a broader point. While the Pol II behaviors are interesting, they could also be explained by the highly crowded and complex nature of RCs that is quite different to the host nucleus. There is DNA replication and packing into virions going on here too, and although the authors mention it they do not address the question of Pol II modification by the virus or viral proteins that recruit Pol II in any meaningful way.

This criticism is a little perplexing, as our goal in this manuscript was to highlight just how the environment of RCs differ from the rest of the nucleoplasm, and how that affects Pol II recruitment. Namely, we show that the high accessibility of the viral genome leads to a number of different behaviors that we elaborate on in the text. We have strong and convincing evidence that molecular crowding is not of particular importance in these behaviors, as it does not change Pol II’s ability to diffuse within or exchange between RCs, shown using multiple different techniques. It is well known that RNA Polymerase II is aberrantly phosphorylated after infection (Fraser and Rice, 2005; Rice et al., 1995), but previous work has shown that this feature does not affect Pol II recruitment to RCs, nor is it absolutely required for productive infection in a tissue culture system. As stated above, we believe we have sufficient evidence that the proteins that are known to interact with Pol II are not primarily involved in its bulk recruitment to RCs, and have attempted to clarify this point in the manuscript.

In parts of the text they mention that there are behaviors of Pol II in the nucleus outside of RCs that are like but not exactly like a normal uninfected cell, which may hint that Pol II modification is involved. This should be explored by testing Pol II mutants in some manner.

We were unable to decipher which claims this reviewer would like to see investigated further, nor what sorts of Pol II mutants they believe would address the question. One possibility is that this comment was based on our description of Pol II CTD truncation mutants used in this paper and more fully described in our recently published report (Boehning et al., 2018). We have referenced our previous report and tried to clarify our use of these mutants in this paper.

The authors do everything in U20S cells. Not only are they transformed cell lines, they are "rogues" in the herpesvirus field; one critical factor in this whole process is ICP0. Yet U20S cells express an as yet unidentified protein that complements the phenotype of an ICP0 mutant and as the only cells known to have this function, they are widely used to grow ICP0 mutant viruses. It would be important to show that this behavior is not an oddity of the unusual choice of cell line in these studies.

It is true that U2OS cells are able to complement ICP0-null HSV1 mutants. Recent work has shown this is a deficiency in the U2OS cGAS/STING innate immune response to foreign DNA (Deschamps and Kalamvoki, 2017), a silencing step that the virus must evade prior to the induction of viral replication and formation of RCs. Given that we focus our study to the events occurring after RC formation, we do not anticipate that our choice of cell line greatly affects our results. Since these cells still produce RCs and recruit RNA Pol II in a manner indistinguishable from what is observed in other published cell lines, and given that we are ultimately interesting in using HSV1 infection as a model system to demonstrate a broader underlying phenomenon of DNA driven compartment formation, we do not see further insight being gained by repeating these experiments in a different cell type since RCs have been demonstrated by many labs to be similar in many cell systems.

There are also issues with the imaging approaches used:1) Representative tracking data (raw images next to overlays with tracks) should be provided for all spaSPT datasets in the paper (infection time course, PAA, pol II inhibitors), movies and figures, in both whole nuclei and enlarged to within RCs. This is necessary for visual confirmation that tracking is accurate within the RCs.

We appreciate the reviewer’s point, as the devil is in the details regarding how masks are generated and further analysis performed. With this said, there are over 3000 individual image files (either or SPT or FISH) which have been annotated, representing multiple terabytes of data, and inclusion of these movies/figure would come at the expense of other supplemental information, as we have reached the maximum allowable supplemental figures. However, the raw data is all available on a data-sharing server with a dedicated DOI number, and we have now added more examples of the celllevel trajectories, including an expanded panel in Figure 2.

Accuracy of tracking in crowded areas should be highlighted specifically (especially if mean localizations per frame increases above 1).

For our spaSPT tracking technique, individual movies average ~0.5 particles per frame to avoid tracking errors, although more recent work from our lab suggests that accurate tracking is possible with might higher densities provided the probability for very long displacements (>800 nm) is sufficiently low.

2) What value ranges are used for DBOUND and DFREE in each dataset? Methods indicate that these ranges overlap, is this correct/appropriate?

We thank the reviewer for their careful reading of this section, which contains a typographical error. The correct values for the bounds used in all conditions are Dfree = [0.5, 25] µm^2^/sec, Dbound = [0.0001, 0.08] µm^2^/sec, FractionBound = [0, 1]. We apologize for any confusion this may have caused, and have corrected this error. For a comprehensive explanation of our methodology and parameter selection process, we refer the reviewer to our recent publication describing SpotOn (Hansen et al. 2018). All methods here adhere to the standards described there. We will stress this point in the manuscript and include relevant quality controls performed on our datasets demonstrating the statistical value of our results.

Why does the DBOUND limit exceed 0.08 µm^2^/sec? Does this indicate viral genomes are unusually dynamic?

This error has been corrected, and the upper limit of the Dbound for fitting is indeed 0.08 µm^2^/sec.

Are these ranges changed between samples within each dataset?

The same ranges are used for all data sets.

If not, why does mean DBOUND change so significantly in Figure S4?

The diffusion of chromatin between consecutive frames is much smaller than the localization accuracy of a single molecule, adding some error to the exact value of Dbound. It is important to keep in mind that a difference of 0.05 µm^2^/sec is not a particularly large difference, and that it only looks large in the graph because of the scale used. We do not interpret these values farther than to say that for all conditions tested they fall well below the diffusion rate of anything aside from chromatin. We have tried to clarify this in the manuscript.

3) If any image acquisition or SPT data modelling values are changed between different samples in each dataset, how comparable are the resulting models? It is necessary to address this question of two-state model interoperability specifically.

No acquisition or modeling parameters are changed between any of the data sets, and as such they are immediately comparable between treatment conditions and time points.

Further, it is important to perform true biological replicates (not just model resampling) to see the real error in the estimation of the mean from these measurements.

All data sets were collected over at least two, but typically four, different imaging sessions, on as many different days. We disagree with the assumption that the day-to-day variability assayed by grouping into “biological replicates” is necessarily more informative or more useful than the cell-to-cell variability that our random resampling approach provides, although we are happy to repeat the analysis treating each of the biological replicates separately. Grouping many independent recordings from different cells into “biological replicates” often gives a false impression of stability and accuracy in the assay since it fails to represent accurately the cell to cell variability. The other important factor to consider is that Spot-On model fitting is more robust as the amount of data is increased. Because of the sparsity of data in RCs, particularly at early times in infection, we sought to use the bootstrapping approach in order to minimize the error in the estimate that is derived from the fitting. Previous work has shown that the mean estimate converges quite quickly, but any individual iteration may be somewhat divergent. We chose to perform bootstrap resampling with 15 cells because this number of cells guaranteed a minimum of 1000 trajectories in the fitting, but would allow us to iterate many times.

Still we appreciate the reviewer’s concern, and so we compared the results of grouping cells into biological replicates as opposed to pooling and randomly subsampling them, and have included this data as a supplemental figure. In summary, either approach yielded results and conclusions that are indistinguishable, within the error measurement of the assay. These results confirm that there are indeed cell-to-cell sources of variability, as well as infection-to-infection, a result that is probably not particularly surprising. We appreciate the reviewer’s comment, and have chosen to include these data as an example of the many different potential sources of error that can arise within single-cell data.

4) Extending this model of non-specific DNA binding to other DNA binding proteins based solely on the data provided in Figure 6 is a stretch. To make such claims it would be important to determine if this holds within RCs for TetR-Halo using spaSPT, and this should be determined for several DNA binding proteins and in contexts outside of HSV1 RCs.

We appreciated the reviewer’s point regarding the figure, and have performed additional experiments to support this conclusion, as shown in the revised Figure 6 and related supplemental figures. We find that the same result when looking at cells expressing the Lac repressor, another bacterial transcription factor which lacks binding canonical binding sites in either virus or human genome. We also established a cell line stably expressing TetR-HaloTag in order to perform SPT on TetR in the context of infection, and find that the same principles we observed with Pol II hold with TetR.

All in all, while the imaging is impressive the conclusions are poorly supported and casually dismiss many alternatives, there is a lack of mechanism here and the advances over prior studies (even if they used more conventional methods) in terms of our understanding of viral genome chromatin state are limited.Minor Points:It would be helpful if the authors referred to viral proteins in a consistent manner. A key protein, ICP0 is referred to as RL2 in the text and figures but ICP0 in the table. This gets confusing.

We have made this change.

If Pol II is proposed to randomly explore "naked" viral DNA, it would be helpful if the authors discussed how the virus might accomplish such a carefully coordinated kinetic gene expression program under such circumstances.

Our discussion centers on the general recruitment of Pol II to RCs, rather than the specific regulation of different kinetic classes of HSV1 genes. We do not find any reason to doubt or modify much of the current consensus regarding the activation of Immediate Early and Early genes, and the process that licenses Late gene transcription. We do briefly comment in the Discussion on the fact that the increase in promiscuous Pol II binding may help explain how so much Pol II loading and transcription occurs at such weak promoter sequences. However, because our focus is to understand the mechanism of general recruitment to RCs rather than gene-specific phenomena, we believe that further commentary would go beyond the scope of our study.

While immunofluorescence is used to determine viral titer in the methods, production of ICP4 does not mean that each of these cells is actually productively infected as some infections can abort. It would be important to show how the titer obtained from determining plaque forming units compares with this measurement, or additionally confirm that the ICP4 positive cells go on to make late proteins also.

Indeed, virus titers were determined using both the standard plaque assay, and additionally using IF. We found that the plaque assay underestimates the number of infected cells by ~20% for wild-type KOS and therefore very much agrees with the reviewer’s comment.

Overall the paper has many typos, "cit" where references should be and is sloppily formatted.

We have made these corrections, where relevant.

Reviewer #2:

In this manuscript by McSwiggen et al., the authors investigate the compartmentalization of HSV1 in the nucleus during lytic infection. They specifically test the hypothesis that the local concentration of viral DNA and transcription machinery constitutes a liquid-liquid phase separated (LLPS) region in the cell as described in a number of recent publications. They carry out the standard LLPS experiments and characterizations such as determination of the aspect ratio, diffusion in the interior and boundary, identification of intrinsically disordered regions in the protein, and disruption via hexanediol. All of these experiments indicate the macroscopic blob visible in the microscope is indeed not a LLPS as previously defined but rather due to an enrichment of non-specific interactions between RNAP2 and the highly abundant nucleosome-free DNA. Thus, their conclusion is that the immense amount of viral DNA concentrates RNAP2 through weak binding interactions. In biophysical terms, they are describing the difference between avidity and affinity. Given the frenzy surrounding this topic of phase separation, I think this paper which takes a rigorous, unbiased approach to the topic is timely and will likely engage the broad readership of the journal. I recommend publication after a few revisions.Major comments:1) My primary concern is on the issue of nucleosome-free regions. There is a debate in the chromatin field about what 'open' chromatin means, but there are a number of papers which seem to indicate that open chromatin (i.e. by ATAC or DNAse hypersensitivity) does not equal nucleosome free (here is one example: PMID 29126175). I suggest the authors supplement their ATAC-seq studies with an IF assay as described in the cited publications but done in their samples under their conditions. I think it is critical to determine what the histone proteins look like in the RC.

We thank the reviewer for this suggestion, and have now included this experiment as a panel in Figure 4.

2) Discussion of phase condensates or liquid-liquid phase separation has reached a fever pitch, and there are likely differences between p-podies, stress granules, heterochromatin, etc. However, the closest analogy I see to this work on transcriptional activation is the nucleolus, which consists of many copies of the ribosomal genes but is transcribed by Pol1, which doesn't have a CTD. I suggest the authors make a direct, detailed comparison of their data to the phase separation work on the nucleolus. For example, I didn't notice 'fusion' of these HSV1 RCs, and this assay has always been one of the most direct measures of LLPS.

Overall, this is an interesting point. To be clear, we do observe some fusion events (see Figure 1B for two examples; and see the related supplemental movies). The comparison with nucleolar structures is an interesting one, especially given some of the pioneering nuclear dynamics work showing that nucleolar factors exchange between nucleoli and the nucleoplasm rapidly, similar to what we see with Pol II. Given the richness of the literature surrounding nucleoli, both in their role as LLPS domains and in the dynamics of nucleolar compartments, the length limitations of the manuscript prevent the rich discussion that this topic deserves. Still, we have attempted to highlight the comparison in the updated Discussion section.

3) Related to this point, it could be that phase separation as postulated in other instances of transcriptional activation is due to the presence of nucleosomes and disordered histone tails, which would provide the crowding agent. If one waits long enough in the infection cycle, does the viral genome become chromatinized, and do the diffusion properties of RNAP2 change?

The reviewer raises an interesting question, which is unfortunately quite difficult to answer since our cells lyse after a few hours releasing their viral load. To our knowledge, there is no way to force chromatinization of the viral genome after lytic infection has begun as RCs are formed. Latency is typically established in neuronal cell types, and while these latent genomes are chromatinized, they exist as single episomes so it is not clear that they would share any of the behaviors of lytic, replicating viral DNA.

Minor comment:1) Include the positive control for drug efficacy in Figure 3.

We have made this change in the new version of the manuscript.

Reviewer #3:

McSwiggen et al. the authors have attempted here to access potential mechanisms by which the Herpes Simplex Virus 1 generates so called replication compartments (RC) within the nuclei of cells in which RNA polymerase II (Pol II) and a number of other proteins are recruited in order to hijack transcriptional machinery to drive expression of viral genes. Following on recent arguments that Pol II may form transcriptional foci via liquid-liquid phase separation (LLPS) and that this phase separation is driven by weak interactions with a Pol II subunit C-terminal domain (CTD) intrinsically disordered heptad repeats, the authors set out to test the hypothesis that Pol II is recruited to RCs through interactions between its CTD and other IDR-containing proteins within the RC. Importantly, the key experiment the authors performed was to sequentially delete heptad repeats and test whether Pol II is no longer recruited to RCs, as one would expect if the CTD heptad repeats are essential to LLPS. They clearly demonstrate that this is not the case. Furthermore, they demonstrate that, although the kinetic exchange of Pol II and fusion of RC are consistent with LLPS, other behavior, for example sensitivity of Pol II focus formation to 1,6hexanediol is not consistent with LLPS. Some further analyses of Pol II dynamics including measurements of jump-length frequencies and sizes and FISH analyses of the viral RNA DNA the authors argue that the formation of RCs is in fact driven by non-specific interactions of Pol II with the viral DNA, which is more open than other chromatin and therefore acts as a sort of sink for diffusion of other transcriptional proteins to enhance expression of viral genes.The study is a very well performed and sophisticated analysis of the problem and based on the assumptions of the central hypothesis, I'd have to conclude that they have proven their point.

We greatly thank the reviewer for this comment.

There is, however, a problem and it's easy to address. First of all, I don't dispute anything that the authors claim about Pol II binding to the viral DNA, but I do not agree that they've excluded LLPS as essential to instigating the Pol II recruitment. The problem rests here: the authors assume LLPS is important to partitioning of Pol II to RCs, it must be driven by interactions of the Pol II CTD with the intrinsically disordered domains of viral proteins. It is possible that LLPS of the viral proteins precedes and is essential to Pol II recruitment; albeit for reasons that are not necessarily clear. The fact that some of the experiments to test for LLPS of the RCs seem negative, I would argue that this is an over-interpretation of these experiments.

Indeed, our initial expectation regarding the recruitment of Pol II to RCs would have been a CTD-mediated process, since we had observed such CTD interactions both in vitro and in uninfected cells as reported in our recent paper (Boehning et al., 2018). However, the results we obtained with HSV1 RCs clearly show that even in very early stages of RC formation Pol II is not being recruited via a CTD mediated mechanism but rather by DNA avidity as succinctly articulated by reviewer #2. So it seems evident that a transition occurs early in the infection cycle. However, we cannot and do not claim to rule out the possibility that the CTD of RNA polymerase II is important during, for example, transcription of IE and Early genes prior to RC formation. And certainly, we are not challenging the notion that the Pol II CTD likely plays a role in hub formation in uninfected cells and perhaps also in the very early stages of viral gene transcription. However, late gene expression, after onset of DNA replication and during formation of RCs is a different story which we believe these studies reveal an elegant and parsimonious mechanism for usurping Pol II from the host chromosome to the naked viral genome.

Our conclusion that Pol II is not accumulating in RCs due to an LLPS process is based on our dynamic measurements of Pol II. In particular, we find that Pol II freely exchanges between RCs (Figure 2F), experiences no barrier to enter or exit RCs (Figure 2G), displays no change in diffusion coefficient (Figure 2E), and shows more than two orders of magnitude in variance of its local density within RCs. None of these observed features would compel one to postulate that the Pol II CTD would mediate these behaviors.

New experiments using other IDRs support our previous findings, showing that these IDRs do not become enriched in RCs despite their propensity to interact with the CTD (Figure 1—figure supplement 1) (Chong et al., 2018). It is unfortunate that 1,6-hexanediol treatment has become such an “acid test” in the burgeoning LLPS field, as it is a very harsh and nonspecific way to address “weak hydrophobic interactions”. Our goal in using 1,6-hexanediol was primarily to demonstrate that we have followed all of the “standard” tests for LLPS, and to show where our system deviates from the current literature. Our unexpected findings also offer a rationale as to why we might suspect that some mechanism other than phase separation could be occurring, but we refrain from interpreting further than this. We have emphasized this better in the text.

In fact the time dependent decreases in mobile fraction of Pol II measured in FRAP experiments are perfectly consistent with the "aging" that has been described for other bodies demonstrated to form by LLPS where the configurations of the network of interactions within a condensate go from loosely associated, relatively low-valency to extended hydrogen bind networks interactions. This change of state is, in fact, common to LLPS-generated bodies, not an exception.

We agree that “aging” can be a common property of LLPS-bodies, but believe that our data clearly rule out aging in this case. Suppose Pol II aging took place in viral RCs. Then one would expect the “apparent k_off_” to decrease or get slower as a function of time (hpi). The clearest measurement of the k_off_ comes from the FLIP measurements (Figure 2F; Figure 2—figure supplement 2). The FLIP data show no difference between infected and uninfected cells and no change in “apparent k_off_” with increasing hpi. Therefore, since the “apparent k_off_” does not change as would be expected for LLPS aging, we feel confident in ruling out LLPS aging for Pol2 RCs.

Regarding FRAP, the recovery time is a complicated function of the chromatin associated fraction, the k_on_, the k_off_, whether diffusion is Brownian and more. Our data clearly show non-Brownian diffusion (Figure 5D-F), and a large change in the chromatin associated fraction (Figure 3D). When taken together, our FRAP and FLIP data show that it is these factors and NOT a change in k_off_ that is responsible for the lower FRAP recovery in Figure 1F and 3H.

So here's what the authors need to do. If you want to exclude LLPS as essential to formation of RCs, start with the hypothesis that the initial step of RC for is LLPS by the viral proteins that they identified, including UL49, RL2 and UL54.

As stated before, we cannot exclude CTD interactions as a potential mechanism very early in the viral infection process, since none of our experiments were directed at analyzing these immediate early or early gene events. We have rephrased the text to clarify this point, and not distract the reader from the main findings that are squarely aimed at the massive recruitment and enrichment of Pol II in RCs following their formation.

Then, do the following experiments:1) Delete the low complexity domains of all combinations of the low complexity domains of these proteins, from individual to pairs to all of the domains, and test whether the viral proteins AND Pol II partition to RCs.2) Next: To exclude the possibility that the low complexity domains are simply forming essential proteinprotein interactions, swap the low complexity domains between the three viral proteins, excluding the individual domains and repeat the experiments in 1.3) Repeat 2, but replace individual domains with another low complexity domain of similar composition and length but completely different origin and test whether RCs form.4) Finally, in each of these cases, test for expression of the viral genes to determine whether transcription is normal.

Reviewer #3 suggests an extensive set of experiments that are logical extensions if we wished to entirely rule out the possibility of LLPS in all steps leading up to the formation of RCs. We’ve addressed these comments above; but in addition, the experiments that are proposed are technically unfeasible for multiple reasons. The first is that many viral proteins perform multiple roles throughout infection. The viral protein ICP27 (gene UL54), for example, has documented roles in gene-specific transactivation/repression, mRNA export, 3’ end processing, and global repression of splicing. While many mutant viruses have been generated by deleting regions of ICP27, it can often be very challenging to identify what is that direct cause of the mutant phenotype. As an example, see our response to reviewer 1 regarding the n504 and n406 mutants. The second major issue is that the HSV genome is very densely packed, often with multiple overlapping ORFs occupying the same sequence space. Again, this makes large deletions and domain swapping nearly impossible because it then becomes unclear what are primary and what are secondary effects of the mutation. Lastly, HSV1 is not a genome that can be readily edited, and successfully incorporating a single edit into a viral mutant can take months. The scope of the experiments proposed would take years to complete, and at the end still may not accomplish the goal of determining what role these low complexity domains play. Indeed, the existence of IDRs in a number of HSV1 genes very likely indicates that some IDR-driven protein:protein interactions are undoubtedly taking place. But clearly, whatever these are doing, they are not affecting Pol II recruitment to RCs.

Nevertheless, we appreciate the importance of this point, and have performed a set of experiments that we hope at least obliquely address the spirit of the proposed experiments. Specifically, we took the low complexity IDRs from the FET family proteins FUS, EWS, and Taf15 because these have been shown to interact with the Pol II CTD in a phase-separation/hub formation mechanism (Chong et al., 2018). We transfected cells with HaloTag-fused to these IDRs, and subsequently infected these cells. Despite their tendency for homo- and heterotypic protein:protein interactions with themselves and with the Pol II CTD, these proteins are not enriched in RCs, nor do these interactions appear to “outcompete” whatever interactions are driving Pol II into RCs. Meanwhile, we have evidence now for two nontargeting DNA-binding proteins (LacI and TetR) that are enriched. We believe that this is very strong evidence that protein:DNA interactions rather than protein:protein interactions are the drivers for Pol II recruitment to the domain.

If none of these make any difference than I will concede that the authors original hypothesis is likely supported. If not, the authors can merely embrace the notion that LLPS is happening and is essential. You can't lose one way of the other.

We thank this reviewer for their general support of the manuscript. However, phrased this way, the reviewer has erected a false dilemma. As we mention above, we cannot conclusively say that LLPS does not occur in the lifecycle of HSV1 or during early events leading to the formation of RCs. However, we dispute the suggestion that LLPS, even if it is occurring between other HSV1 proteins, is necessarily essential to their function. To our knowledge, no group has yet demonstrated a case where LLPS is essential to the functioning of transcriptional regulatory systems. Instead, many IDRs play a role in the formation of transient local high concentration hubs, but there is little or no evidence that these functional hubs actually form LLPS condensates. Indeed, this is a major point of discussion in the field at large.

In either case, we do not see how conceding that some LLPS may occur between some viral proteins during some stages of the viral life cycle changes the interpretation of our data, which is that these forces are not what govern Pol II recruitment into RCs, and that Pol II does not “experience” the defining constraints of phase separation even if it does occur. We have updated the manuscript to emphasize this point more forcefully.

[Editors' note: the authors’ responses to the re-review process at another journal follow.]

Summary:

After implementing the changes and suggestions from the reviewers’ first round of comments, the responses were generally even more positive. All three reviewers accept the quality of the data we presented, with the newly incorporated changes, and the remaining comments largely center on interpretation of the data. Reviewer #2 suggests the manuscript should be published as revised, and even goes on the recognize the bias that Reviewer #3 shows in their response. Reviewer #1’s criticisms to the revised manuscript focus on the fact that HSV1 expresses proteins involved in transcription activation. As is discussed in more detail below, we contend that even if these proteins are involved in Pol II recruitment, we show that the mechanism we propose applies broadly not just to Pol II, but to any other viral complex that has the capability to bind DNA. Reviewer 3 again telegraphs their own biases, ignoring key data in the manuscript as well as their own previous comments in order to maintain their assertion that HSV1 RC formation is yet another form of phase separation.

Reviewer #1:

There is no disputing that the authors use elegant approaches that demonstrate non-specific binding or random exploration as a means of PolII retention at RCs, making this an important report. But serious concerns remain about the overinterpretation of data, weakly supported at points and with particular disregard to the underlying biology of infection, along with one-sided views that should at the very least be addressed up front in the manuscript to improve fairness and overall readability – unless I am completely missing something, in which case I am happy to be corrected.Major Points:The rebuttal often seems at odds with what is in the actual paper:A key point of contention is the idea that viral proteins are simply resistant to 1,6-HD and mediate recruitment of PolII to viral DNA. I actually don't have much of a problem with the novelty of random exploration mediated by viral proteins, it is still novel I think? Yet the authors stubbornly refuse to do any experiments beyond imaging to test this idea and refuse to discuss the notion.

The reviewer’s initial comments highlight an important aspect of our model that we have clearly not communicated well enough, and will try harder to improve in future versions of the text. Imagine that rather than doing these SPT experiments with RPB1, the Pol II catalytic subunit, we had instead used an accessory subunit such as RPB3. We have fairly strong evidence to suggest that in this hypothetical scenario we would obtain the same values and behaviors by spaSPT and FRAP. Assuming this were the case, we would propose the same model: That nonspecific binding of Pol II, as a protein complex, to the viral DNA drives RC accumulation. The fact that the particular subunit we are labeling may or may not itself have ability to bind DNA is immaterial, because we are considering Pol II and its behavior as a complex. We know of no experimental technique that can show in cells that it is specifically the RPB1 catalytic subunit that is making nonspecific contact with the DNA, and so in our model we don’t make so specific of a statement, but appreciate that this could be more clearly specified.

We know from previous literature that many viral proteins interact with many host proteins, including Pol II. We cannot fully exclude the possibility that a viral protein binds Pol II forming a new complex, and that this is the interaction that drives nonspecific binding to the viral DNA. We disfavor this model for the reasons that are detailed below. More importantly, our model is that any DNA-binding protein, viral or human in origin, will be subjected to the same mechanism of nonspecific binding and that this is occurring contemporaneously with all of the other protein-specific mechanisms that have previously been elucidated.

We will make this point more clearly and explicitly in the manuscript going forward.

For example, the rebuttal states:"As the reviewer points out, we cannot say for sure that 1,6-HD disrupts ICP27-Pol II interactions, but taken together with the above arguments, we believe that we have sufficient evidence to disfavor ICP8/ICP27 interactions as a major contributor to bulk recruitment of Pol II to RCs". Cases of viral proteins mimicking the function of cellular proteins yet being both structurally dissimilar and resistant to conventional inhibitors of their host counterpart abound. HSV-1 actually encodes a classic example, the Us3 kinase that mimics Akt but is resistant to Akt inhibitors.

We thank the reviewer for this perspective, and would offer a few points in response. Firstly, the reviewer should think of the use of 1,6-hexanediol as less comparable to inhibitor compounds like Akt inhibitors, or the drugs we use later in the manuscript Flavopiridol and Triptolide, and more like treatment with a hypotonic buffer. That is to say, 1,6-hexanediol is supposed to inhibit the weak van der Waals forces that are thought to drive interactions between just about every unstructured protein domain. At a concentration of 10%, this treatment is strong enough to melt nuclear speckles, PML bodies, Cajal bodies, transcription “factories”, and transcription factor-Pol II interactions, to name a few. We note in the manuscript that you can see the results of such harsh treatment in the complete disruption of nuclear morphology in the treated cells. As harsh as this treatment is, we use it in Figure 1 because it has become something of a “standard” assay when looking at phase separation in cells.

With the above comments in mind, we do not know whether 1,6-HD specifically impacts ICP27-Pol II interactions. If the interactions are predominantly the result of weak and hydrophobic interactions, it very well might, but if the binding is due to ionic interactions then this may not be the case. In either case, probing this specific interaction wasn’t the goal of the 1,6-HD experiments. Rather, we were applying a commonly used assay in the field of phase separation to test whether RCs exhibit the same behavior (i.e. being driven by weak hydrophobic interactions) as other phase-separated compartments—which they do not.

The latter point disfavoring IC8/ICP27 is only supported by data in Figure S5 that not only contradicts findings in the field, according to the authors interpretation, but to me the n406 mutant that is fully defective in Pol II binding seems to exhibit pretty severe defects. Only one cell is shown and only 10 are used for FRAP analysis, while 30 or more are used for WT. While I understand the point is that these small RCs recruit Pol II and exhibit similar FRAP profiles, there is limited analysis here to truly show what is going on particularly given how it is viewed as refuting findings from well-respected virologists. How much viral DNA is in these cells (is this limited exploration of large amounts of viral DNA for example, which would be counter to their model), and what is happening in the majority of cells that they mention are defective? Quantitation is limited: What is the proportion of the total population that exhibit these behaviors, and are the ones shown simply "escapees" that struggle to form RCs?

It is absolutely the case that these ICP27 mutants are replication deficient. Given this protein’s central role in not only activation of immediate early and early gene transcription, but also RNA processing and export, it shouldn’t be surprising that even small modulations of this protein’s potency would affect the virus’ ability to replicate. However retarded in their growth they are, RCs do form (seen both by presence of ICP4 and decrease in DAPI staining) and, more importantly, they still recruit Pol II. We have not measured the DNA content of n406 RCs, as this is not a trivial undertaking. Still, it is telling that the RCs that form are larger than the pre-replicative compartments seen in PAA-inhibited WT infections (Figure 4 and S6), and have FRAP recovery profiles more consistent with WT infections than they do with PAA-inhibited samples (this data will be included in Figure 3—figure supplement 1 going forward).

Regarding the numbers of cells measured by FRAP, 30 cells is overkill for the type of analysis we are doing (i.e. qualitative comparison of recovery profile). We had started making many more measurements because we had initially planned to fit the data to binding/diffusion models, and only after collecting all of the data found that RCs do not satisfy all of the assumptions to use FRAP data in this quantitative manner. With this in mind, we collected data for a more reasonable number of cells for all of the mutants and drug treatments.

Regarding our finding contradicting published literature, we would invite the reviewer, if they have not already done so, to read the paper in question (Dai-Ju, J.Q., Li, L., Johnson, L.A., and Sandri-Goldin, R.M. (2006). ICP27 Interacts with the C-Terminal Domain of RNA Polymerase II and Facilitates Its Recruitment to Herpes Simplex Virus 1 Transcription Sites, Where It Undergoes Proteasomal Degradation during Infection. J. Virol. *80*, 3567– 3581. DOI: 10.1128/JVI.80.7.3567-3581.2006) to judge whether they agree with the authors’ interpretation of the data presented in that paper. A close rereading, specifically of Figure 4D and the associated text, suggest that the authors actually see something very similar to what we observe here – that Pol II is indeed enriched in RCs. Thus our data are mostly in agreement with the data they show in this figure, however the addition of FRAP experiments have caused us to come to a different conclusion than what they suggest.

In addition, other viral proteins could be mediating this. As the authors themselves state in the rebuttal, "a number of viral proteins interact with Pol II or other members of the core Pre-Initiation Complex. This includes the viral proteins VP16, ICP4, ICP8, ICP27, and ICP22 among others". Claims that PolII diffusion is not affected don't really support there being no role for viral proteins – if they are small or transiently interact there might not be a discernable change in PolII behavior. It seems inappropriate to simply ignore these possibilities to claim they cannot reproduce others’ findings and one-sidedly push their model of "naked" DNA.

The reviewer is correct that a sufficiently small or transient interaction with Pol II may go undetected by SPT. It is possible that some other viral protein besides the ICP27/ICP8 complex might interact directly with Pol II and thereby facilitate recruitment, though there is little biochemical evidence to support this. In a hypothetical scenario where ICP27/ICP8 or another viral protein is involved, this doesn’t really solve the question of how Pol II (or any other protein) is specifically enriched in RCs, but rather only abstracts it one level. How, then, does viral factor X get to RCs? Especially given that most HSV1 proteins, including ICP27 and ICP8, localize to RCs while lacking sequence-specific DNA binding motifs that would direct them to the viral genome over the host genome, and where we have convincingly demonstrated there is no border enclosing RCs to constrain host or viral proteins within RCs, invoking interaction with viral proteins doesn’t improve our understanding of the underlying mechanism. What we rather hope we have demonstrated is that the virus is capitalizing on its highly accessible genome (more on this point below) to affect the recruitment of many proteins – both host and viral in origin. We favor a model where Pol II is directly interacting with the viral DNA because of the experiments outlined above and because it doesn’t require invoking additional host or viral factors, but we will make sure to clarify in the text what our assumptions are in this model, and what are its limitations.

Indeed, key section statements like viral DNA being far more accessible than host DNA require clarification as it implies naked DNA, and the broader inference throughout is that the DNA is naked even when not explicitly stated.Oddly, the rebuttal says "We make no claims that these drugs work against any viral proteins" and "We had no intention to suggest that Pol II only interacts with the viral DNA". Yet throughout the paper statements to that effect are made e.g.; "Pol II recruitment occurs predominantly through transient, nonspecific binding of Pol II to naked viral DNA." Or: "Within RCs, many copies of the unprotected HSV1 DNA are present" etc. The reality is HSV-1 encodes many non-specific DNA binding proteins, transcription factors etc. and DNA is engaged in replication, transcription and packaging and is unlikely to be "naked". Perhaps regions are exposed, but the authors should be specific. They also claim "competition" with host chromatin? The virus affects host transcription, so it's not a simple competition as far as I am aware. Again, I don't have a problem with the data, this is all about statements that don't consider the biology of infection, which should be discussed. While this paper has interesting implications for cell biology, it lacks context for what is actually happening during infection.

ATAC-seq is a measurement of DNA accessibility (Klemm et al., 2019), of which it has been shown that the absence of nucleosomes is of primary importance relative to any other class of DNA binding protein. While naked is a colloquial term, it is one we feel is very fitting for this situation. A person wearing 100-fold less clothes is by any metric going to be considered naked with respect to their peers; and just as with people, it is quite shocking to see any DNA in a eukaryotic nucleus so devoid of its nucleosomal armor. With this said, the reviewer is correct that it would be more accurate to replace “naked” with “non-nucleosomal”, or something to this effect.

As stated above, we believe the most parsimonious model given all of the data that we’ve presented is one in which Pol II is predominantly recruited to RCs through nonspecific interactions with the unprotected viral DNA, and so it seems appropriate for that to be a sentence we use to summarize our results without adding qualifying statements.

Regarding the discussion of competition between host and viral chromatin, while it is absolutely true that HSV1 disrupts transcription of the host genome through a wide variety of mechanisms (phosphorylation defects affecting promoter binding and escape, transcription termination, etc.), they all appear to occur at steps that happen after a polymerase has bound to a PIC. For the binding itself, all the DNA occupying the nucleus is in potential competition for Pol II binding, and based on our model it is clear that the HSV1 DNA is outcompeting the host chromatin for this initial step of DNA binding. We believe our experiments with TetR (and now also LacI) are crucial experiments because they demonstrate that the mechanism we are describing for recruitment of Pol II and other DNA-binding proteins is occurring regardless of all the other potential interesting mechanisms that HSV1 employs, which may be occurring contemporaneously.

Regarding the point about the crowded RC, and rebuttal "This criticism is a little perplexing, as our goal in this manuscript was to highlight just how the environment of RCs differ from the rest of the nucleoplasm, and how that affects Pol II recruitment", the point is that the authors don't consider the nature of this crowded environment where DNA replication and packaging may affect the apparent behavior of Pol II, as it may be continuously kicked off DNA that is being used for purposes other than transcription. The claim is high accessibility of the viral genome leads to a number of different behaviors but is it really accessible, or is Pol II simply struggling to compete for DNA binding here? While histones might be absent, they haven't ruled out other proteins occupying the DNA.

We approached this project expecting the environment of the RC to be different from the rest of the nucleus, and in some respects the reviewer is correct in saying that the environment is different. However, it is not clear exactly how different and how much this contributes to the behavior of macromolecules in the infected nucleus. For example, for all of the factors that we’ve measured by spaSPT, RCs have no effect on the mean diffusion coefficient of any of them. Moreover, the quantitative phase imaging suggests that RCs are, if anything, *less* densely packed with DNA than the surrounding nucleoplasm. We do not discuss this in the text of the paper because we were unable to get satisfactory standards to convert gray values into absolute dry mass, but the fact that RCs appear as dark/black spots in the image instead of bright/white spots like the nucleolus does suggest that there is generally lower dry mass inside of the RC than the surrounding nucleoplasm. If this is truly the case, however, it does not appear to have an effect on how molecules diffuse through the space, which generally underscores how much the field still has to learn or relearn about what diffusion looks like in low Reynolds Number environments like the cell.

The reviewer is quite correct that DNA devoid of nucleosomes is not devoid of protein, but there is a large chasm of difference between the interaction of DNA-binding proteins like replication and transcription factors, and nucleosomes. Measurements taken on the stability of nucleosomes bound to DNA suggest that the average nucleosome will stay bound to DNA for hours after deposition, whereas even the strongest and most stable DNA binders like TATA-binding protein and Cohesin have residence times of minutes (Rhodes et al., 2017; Teves et al., 2018). The viral DNA is likely decorated with many proteins, including replication complexes and transcription complexes, but it is very unlikely that there is anything binding so stably to the viral DNA to appreciably affect Pol II binding.

While it might be incorrect to state tegument proteins are structural, they are components of the virion but many of them are not involved in transcriptional control. If you focus only on transcriptional regulators, which are known, how does this affect the IDR distribution? It is possible many unrelated proteins in the virion have IDRs that skew this analysis.

As with the 1,6-hexanediol experiments, these data are included in the manuscript because this is another “standard practice” in the phase-separation field. The logic appears to be something like: proteins with high intrinsic disorder implies proteins likely to interact through multivalent/hydrophobic interactions which imply phase separation. One of the goals of this paper is to offer a refutation to this particular chain of logic: that the existence of IDRs does not necessarily imply phase separation. To this end, we could just have plotted all viral proteins against the proteins known to undergo phase separation, or subdivided the virus into any number of other categories arbitrarily, and it would not change the conclusion.

With that said, we’re happy to adjust the plot to keep the proteins categorized strictly by their kinetic class. Either way, it doesn’t change our conclusion from the figure (Figure 1E).

The use of LacI and TetR (mislabeled TatR in Figure 6) is a nice approach and supports the notion that random exploration likely happens. However, only a couple of cells are shown and everything is done in infected cells. Do these proteins form aggregates that generate PolII concentrations in uninfected cells? Viral proteins are also highly promiscuous protein binders and could again be involved. Although unlikely, at least show uninfected cells to show specificity and indicate the frequency at which these structures form; it was not clear in figure legends or methods.

Neither LacI or TetR form aggregates in uninfected cells (Chong et al., 2018; Normanno et al., 2015), but we’re happy to include images from uninfected cells as well. It seems to us very unlikely that the virus would be capable of binding and recruiting both proteins, bacterial transcription factors which the virus would never have encountered over the course of its evolution. Additionally, Figure S2 shows that RCs do not accumulate the activation domains of FUS, EWS, or TAF15, nor HaloTag alone. We will also include an estimate of how often TetR and LacI are recruited to RCs.

Why is it that "SPT data for TetR-Halo were not well fit by the two state model in Spot-On, however a qualitative assessment can be made from the CDF curves".

There are a number of potential reasons that certain proteins, for one reason or another, are not well fitted by Spot-On. One of the assumptions that Spot-On makes in fitting is that state transitions (bound -> free, or free-> bound) are assumed not to occur on a fast enough timescale that they appear in the trajectories. Hansen et al., 2018, Figure 3-supplemental figure 10 shows that the model fails and fits the data poorly if state transitions occur fast enough to be non-negligible. Given what we know about how TetR binds DNA, it is quite possible that this is the case. Despite this, it is the cumulative distribution function that Spot-On is using to fit the model, so gross changes in the shape of the curve like those we see for TetR can still be interpreted to indicate a shift towards binding within the RCs, even if we can’t precisely measure what fraction of molecules are bound using existing tools.

To be clear, I am not arguing against the novelty of the findings or the paper overall, but I don't understand why the possible role of viral proteins is not more rigorously addressed and the insistence on implying naked DNA is so important here. There seems to be a grossly one-sided interpretation of data that needs to be more balanced, and a fairer acknowledgment of what has not been ruled out in this study. Again, maybe I am completely missing the point here and I'm happy to be schooled otherwise.

We thank the reviewer for their careful rereading of our revised manuscript, and hope that the above comments address their lingering concerns. Ultimately, we are not trying to subvert decades of research on HSV1 in favor of our findings, but rather are highlighting a new mechanism that we believe helps support previous work. This mechanism is not unique to Pol II (or TetR and LacI, for that matter), but applies generally.

Reviewer 2:

The authors have satisfactorily addressed my concerns.I would also like to concur with their statement to Reviewer #3: "To our knowledge, no group has yet demonstrated a case where LLPS is essential to the functioning of transcriptional regulatory systems. Instead, many IDRs play a role the formation of transient local high concentration hubs, but there is little or no evidence that these functional hubs actually form LLPS condensates. Indeed, this is a major point of discussion in the field at large." Indeed. One of the main reasons it that active genes appear as diffractionlimited spots in most cases, ruling out interrogation of internal composition which is essential to a definitive demonstration of LLPS. It is for this reason that studies are being carried out on superenhancers and viral replication compartments: the diffusion across the boundary is a key experiment in the opinion of this referee and can only be done on macroscopic blobs. Moreover, the fact that the authors demonstrate that the phenomena can be explained with a kinetic description, independent of phase separation, is a strong argument.

Reviewer 3:

McSwiggen et al. have made a valiant attempt to respond to my criticisms. I remain, however, unconvinced that the RCs are the result of some alternative type of organization to phase-separated condensates. I do understand their claim that it is not the Pol II CTD interactions with low complexity domain-containing viral proteins that leads to Pol II sequestration within RCs, the anisotropic dynamics of Pol II that occurs inside of them. The question is, why would non-specific interactions of Pol III with DNA be any different in other regions of the chromatin with similar nucleosome occupancy? This is not at all clear to me.

As we demonstrated in Figures 4 and 5A, there is a significant difference between the viral DNA and any other site in host chromatin. To our knowledge, there exists no region of the host genome that even comes close to being as depleted of nucleosomes as the viral DNA is in RCs. Because of this, a DNA-binding protein has much less restricted access to DNA once in the RC, thereby facilitating many more nonspecific interactions.

Further, I can appreciate the technical difficulties that the authors describe to performing the experiments that I suggested and I also appreciate them performing alternative experiments of overexpressing proteins that bind to the CTD of Pol II to test whether they affect Pol II sequestration in RCs, but for reasons I describe below, I think that a different interpretation of their results is possible.So how do we get out of this mess? First of all, the authors do say that they cannot rule out that LLPS is occurring early in the process. So why not simply hypothesize that indeed is what's happening? One could argue that all evidence points to this possibility except for the problem of Pol II CTD interactions not being important.

We have provided in the manuscript, and highlighted in the text, multiple lines of evidence to suggest that LLPS is not occurring. First, the RCs are not dissipating even when exposed to high concentrations of 1,6-hexanediol (Figure 1H). Secondly, SPT analysis of Pol II shows that there is no change in diffusion coefficient upon entering the RC, suggesting no change in viscosity as LLPS would predict (Figure 2E), and the Tet repressor shows no penalty for crossing into or out of RCs (Figures 2F and G, Figure S7C). Third, the addition of a DNA-binding domain (Lac repressor or Tet repressor, Figure 6) is sufficient to drive recruitment of a protein to RCs, whereas addition of protein domains known to undergo LLPS is not (Figure S1). Fourth, PALM data from both the viral DNA (Figure 5A) and RNA Pol II (Figure 5G andH) show that within the RC these molecules are not randomly distributed as one would predict from an LLPS model, but rather show clustering below the scale of the RC. Taken together we believe this strongly disfavors any potential model relying on LLPS. Furthermore, it is somewhat problematic that invoking LLPS has now mysteriously, in the mind of this reviewer, become the null hypothesis rather than a model that requires a higher bar for assignment particularly in vivo.

This also explains why expressing FUS, etc., have no effect. So what? Let's assume that RCs are condensates in which Pol II can enter and exit the chromatin-protein meshwork quite freely, making very weak interactions with all molecular species in the condensate.

Let us assume for the moment, as the reviewer suggests, that RCs do indeed represent a new type of condensate that recruits Pol II (and other DNA-binding proteins) selectively. This model would be just as problematic, if not more so, for anyone hoping to understand the functional role of phase condensates. As stated above, we have found through a battery of assays that being in the RC gives no better understanding of how a given molecule will behave.

An important consideration, one that authors don't account for, is phase separation of chromatin itself. It has long been appreciated that chromatin phase separates, euchromatin from heterochromatin and even different regions of either of these types of chromatin from each other. Importantly, there is very recent compelling evidence that chromatin phase separates dependent on histone complex composition and posttranslational modifications of histones and other protein binding to chromatin (Gibson, et al., http://dx.doi.org/10.1101/523662; Sanulli, et al., http://dx.doi.org/10.1101/473132).

The reviewer offers another red herring regarding the question at hand, both because we show conclusively that there are no histones incorporated into the viral DNA and because neither of the studies referenced demonstrate that the LLPS tendencies shown in vitro translate directly to LLPS inside the cell. In fact, while there is certainly evidence for the role of HP1 and related proteins in the formation of heterochromatin domains in cells in certain contexts, those are a far cry from a system that is marked by absence of the ingredients required to build a liquid heterochromatin domain.

In light of these results, it is interesting to note that the authors observe a paucity of nucleosomes in the viral DNA that is in RCs, consistent with modifications that could increase nucleosome dynamics and resulting in changes in the properties of the viral DNA, changes that might contribute to its phase separation. All of the diagnostic work the authors have done point to RC being a viscoelastic condensate that must be formed in order to sequester Pol II.

We would again remind the reviewer that whatever recruitment mechanisms are at work, they must be acting not only on Pol II, but on many other nuclear factors, including foreign transgenes products like the Tet and Lac repressors. Additionally, Pol II is not sequestered, but remains free to enter and leave the compartment, as well as diffuse within the compartment, without penalty or change in diffusion coefficient. It is truly puzzling that the reviewer chooses to favor a model of RCs as condensates with so much evidence to the contrary.

I will grant you that the field of biomolecular condensates hasn't come up with a strict definition of what these things are and why they wouldn't be simply mistaken for a network of protein-protein interactions.

This single sentence perhaps most clearly embodies our disagreement with the reviewer, as well as a major fault in the field at large. We completely agree that biomolecular condensates remain poorly defined in most contexts, especially when identifying them in vivo. While Reviewer 3 appears to be of the opinion that the label of “viscoelastic condensate” can and should be applied to a system in spite of evidence showing that it differs in some key aspects, we believe that our data underscore the weakness of such an argument. LLPS has very specific physical interpretations and makes key predictions (as Reviewer 3 succinctly lays out below), only some of which are satisfied by HSV1 RCs, and some of which are clearly violated.

At a recent meeting, Rohit Pappu, certainly the leading theorist of the biomolecular condensate field, posed the following set of definitions: "To be a condensate that arises from phase separation, there has to be a saturation concentration threshold above which condensates form and below which condensates dissolve. And because phase separation is a collective phenomenon defined by infinite cooperativity, the interactions that stabilize condensates will be quantifiably non-stoichiometric in nature. These two conditions are necessary for stipulating that a condensate is a phase separation. In addition, a diluent can dissolve the condensate by preferentially interacting with the key regions that are required for forming the condensate.”

While we believe there is still some debate, as the reviewer mentioned in the line above, regarding the exact definition of LLPS condensates, this definition seems as good of a jumping-off point as any. By this definition, we can show that RCs cannot be LLPS condensates simply by examining the Pol II PALM data (Figure 5G). Within each RC, the concentration of Pol II can vary by more than two orders of magnitude, and Pol II shows clustering at all length scales less then 1 µm (Figure 5H). If RCs were undergoing LLPS, then in the concentrated phase (the RC) we would expect Pol II to reach but not exceed the critical concentration. In a system like this, the L(r)-r curve should remain at zero for all length scales because all molecules in the concentrated phase should be at the same local concentration (and thus spatially randomly distributed). Instead we see a range of concentrations of Pol II that dip below the concentration of the dilute phase, and soar way above the mean concentration of the concentrated phase, clearly violating the one tenant that the reviewer has set forth as a key requirement for invoking LLPS.

It would be difficult for the authors to prove the first condition but if RCs are composed of phase separated DNA, it is at least arguable that it forms non-stoichiometric complexes and they do see dissolution by 1,6-hexanediol. A further and simple experiment that the authors could do with 1,6hexanediol is first to titrate the RCs with it and see if they observe a sharp, all or none transition, in which the RCs disperse. They should also show that this is simply reversible by removing the 1,6-hexanediol. If they see a sharp transition instead of a linear degradation of RC structure, it would suggest an infinitely cooperative transition as would be expected for a phase -separated condensate. They should also do a control of the same experiment but using 1,2,3-hexanetriol, which has not been shown to dissolve biomolecular condensates.

We are perplexed by this proposed experiment, as we do not see dissolution by 1,6-hexanediol, even at a very high concentration. It is unclear how much higher one can even titrate the compound before proteins begin to denature.

References:

Boehning, M., Dugast-Darzacq, C., Rankovic, M., Hansen, A.S., Yu, T.-K., Marie-Nelly, H., McSwiggen, D.T., Kokic, G., Dailey, G.M., Cramer, P., et al. (2018). RNA polymerase II clustering through carboxy-terminal domain phase separation. Nat. Struct. Mol. Biol. 316372.

Boissière, S. La, Izeta, A., Malcomber, S., and Hare, P.O. (2004). Compartmentalization of VP16 in Cells Infected with Recombinant Herpes Simplex Virus Expressing VP16-Green Fluorescent Protein Fusion Proteins. J. Virol. *78*, 8002–8014.

Chang, L., Godinez, W.J., Kim, I.-H., Tektonidis, M., de Lanerolle, P., Eils, R., Rohr, K., and Knipe, D.M. (2011). Herpesviral replication compartments move and coalesce at nuclear speckles to enhance export of viral late mRNA. Proc. Natl. Acad. Sci. U. S. A. *108*, E136–E144.

Chong, S., Dugast-Darzacq, C., Liu, Z., Dong, P., Dailey, G.M., Cattoglio, C., Heckert, A., Banala, S., Lavis, L., Darzacq, X., et al. (2018). Imaging dynamic and selective low-complexity domain interactions that control gene transcription. Science *2555*, eaar2555.

Crumpacker, C. (1992). Mechanism of action of foscarnet againist viral polymerases. Am. J. Med. *92*, 3S–7S.

Dai-Ju, J.Q., Li, L., Johnson, L.A., and Sandri-Goldin, R.M. (2006). ICP27 Interacts with the C-Terminal Domain of RNA Polymerase II and Facilitates Its Recruitment to Herpes Simplex Virus 1 Transcription Sites, Where It Undergoes Proteasomal Degradation during Infection. J. Virol. *80*, 3567–3581.

Deschamps, T., and Kalamvoki, M. (2017). Impaired STING Pathway in Human Osteosarcoma U2OS Cells Contributes to the Growth of ICP0-Null Mutant Herpes Simplex Virus. J. Virol. *91*, e00006-17.

Fraser, K.A., and Rice, S.A. (2005). Herpes Simplex Virus Type 1 Infection Leads to Loss of Serine-2 Phosphorylation on the Carboxyl-Terminal Domain of RNA Polymerase II. J. Virol. *79*, 11323–11334.

Hancock, M.H., Cliffe, A.R., Knipe, D.M., and Smiley, J.R. (2010). Herpes Simplex Virus VP16, but Not ICP0, Is Required To Reduce Histone Occupancy and Enhance Histone Acetylation on Viral Genomes in U2OS Osteosarcoma Cells. J. Virol. *84*, 1366–1375.

Klemm, S.L., Shipony, Z., and Greenleaf, W.J. (2019). Chromatin accessibility and the regulatory epigenome. Nat. Rev. Genet. 1.

Lee, J.S., Raja, P., and Knipe, D.M. (2016). Herpesviral ICP0 Protein Promotes Two Waves of Heterochromatin Removal on an Early Viral Promoter during Lytic Infection. MBio *7*, e02007-15.

Normanno, D., Boudarène, L., Dugast-Darzacq, C., Chen, J., Richter, C., Proux, F., Bénichou, O., Voituriez, R., Darzacq, X., and Dahan, M. (2015). Probing the target search of DNA-binding proteins in mammalian cells using TetR as model searcher. Nat. Commun. *6*, 7357.

Rhodes, J.D.P., Haarhuis, J.H.I., Grimm, J.B., Rowland, B.D., Lavis, L.D., and Nasmyth, K.A. (2017). Cohesin Can Remain Associated with Chromosomes during DNA Replication. Cell Rep. *20*, 2749–2755.

Rice, S., Long, M., Lam, V., Schaffer, P., and Spencer, C. (1995). Herpes simplex virus immediate-early protein ICP22 is required for viral modification of host RNA polymerase II and establishment of the normal viral transcription program. J. Virol. *69*, 5550–5559.

Rutkowski, A.J., Erhard, F., L’Hernault, A., Bonfert, T., Schilhabel, M., Crump, C., Rosenstiel, P., Efstathiou, S., Zimmer, R., Friedel, C.C., et al. (2015). Widespread disruption of host transcription termination in HSV-1 infection. Nat. Commun. *6*, 7126.

Scott, E.S., Malcomber, S., and O’Hare, P. (2001). Nuclear Translocation and Activation of the Transcription Factor NFAT Is Blocked by Herpes Simplex Virus Infection Nuclear Translocation and Activation of the Transcription Factor NFAT Is Blocked by Herpes Simplex Virus Infection. J. Virol. *75*, 9955–9965.

Taylor, T.J., McNamee, E.E., Day, C., and Knipe, D.M. (2003). Herpes simplex virus replication compartments can form by coalescence of smaller compartments. Virology *309*, 232–247.

Teves, S.S., An, L., Bhargava-Shah, A., Xie, L., Darzacq, X., and Tjian, R. (2018). A stable mode of bookmarking by TBP recruits RNA Polymerase II to mitotic chromosomes. *eLife 7*.

Wyler, E., Menegatti, J., Franke, V., Kocks, C., Boltengagen, A., Hennig, T., Theil, K., Rutkowski, A., Ferrai, C., Baer, L., et al. (2017). Widespread activation of antisense transcription of the host genome during herpes simplex virus 1 infection. Genome Biol. *18*, 1–19.

Zhou, C., and Knipe, D.M. (2002). Association of herpes simplex virus type 1 ICP8 and ICP27 proteins with cellular RNA polymerase II holoenzyme. J. Virol. *76*, 5893–5904.